# CAMELS-Chem: Augmenting CAMELS (Catchment Attributes and Meteorology for Large-sample Studies) with Atmospheric and Stream Water Chemistry Data

Gary Sterle[1], Julia Perdrial[2,7,] Dustin W. Kincaid[3,7], Kristen Underwood[3,] Donna Rizzo[3,7,] Ijaz Ul Haq[4], Li Li[5], Byung Lee[4], Thomas Adler[2,7,] Hang Wen[6,] Helena Middleton[1] and Adrian Harpold[1]

[1] Department of Natural Resources and Environmental Science, University of Nevada, Reno, USA
[2] Department of Geography and Geosciences, University of Vermont, USA
[3] Department of Civil and Environmental Engineering, University of Vermont, USA
[4] Department of Computer Science, University of Vermont, USA
[5] Department of Civil and Environmental Engineering, Pennsylvania State University, University Park, USA
[6] School of Earth System Science, Tianjin University, Tianjin, China
[7] GUND Institute of the Environment, University of Vermont, USA

*Correspondence to*: Adrian A. Harpold (aharpold@unr.edu), Julia Perdrial (Julia.Perdrial@uvm.edu)

**Abstract.** Large sample datasets are transforming the catchment sciences, but few off-the-shelf stream water chemistry datasets exist with complementary atmospheric deposition, streamflow, meteorology, and catchment physiographic attributes. The existing CAMELS (Catchment Attributes and Meteorology for Large-sample Studies) dataset includes data on topography, climate, streamflow, land cover, soil, and geology across the continental U.S. With CAMELS-Chem, we pair these existing attribute data for 516 catchments with atmospheric deposition data from the National Atmospheric Deposition Program and water chemistry data and instantaneous discharge from U.S. Geological Survey over the period from 1980 through 2018 in a relational database and corresponding dataset. The data include 18 common stream water chemistry constituents: Al, Ca, Cl, dissolved organic carbon, total organic carbon, $HCO_3$, K, Mg, Na, total dissolved N, total organic N, $NO_3$, dissolved oxygen, pH (field and lab), Si, $SO_4$, and water temperature. Annual deposition loads and concentrations include hydrogen, $NH_4$, $NO_3$, total inorganic N, Cl, $SO_4$, Ca, K, Mg, and Na. We demonstrate that CAMELS-Chem water chemistry data is sampled effectively across climate, seasons, and discharge for trend analysis and highlight the coincident sampling of stream constituents for process-based understanding. To motivate use by the larger scientific community across a variety of disciplines, we show examples of how these publicly-available datasets can be applied to trend detection and attribution, biogeochemical process understanding, and new hypothesis generation via data-driven techniques.

# 1 Introduction

Earth surface processes include coupled and complex interactions that involve atmosphere, biosphere, lithosphere and hydrosphere, however, tracking these important dynamics across time, space and disciplines remains a challenge that is, amongst others, related to data availability and connectedness. To address the need for 'balancing breadth with depth' (Gupta et al. 2014, Hubbard et al. 2020), the hydrological sciences have developed large sample size datasets that are of high quality and then made these available to the research community. One of the key advantages of aggregating and harmonizing data into larger sample size datasets is to test how model hypotheses reproduce observed behaviour across variable conditions and sites to reduce the uniqueness of place issues (i.e. individual catchment behavior might not be generalizable to explain a larger scale pattern and vice versa, Gupta et al. (2014) and Hubbard et al. (2020)). As a result, several recent efforts have focused on generating and using datasets across the continental U.S. (CONUS) where observation networks are relatively dense. For example, the Model Parameter Estimation Experiment (MOPEX, Duan et al. (2006)) dataset has been used to detect the effects of shifts from snow to rain on streamflow (Berghuijs et al. 2014) and to better diagnose the cause of catchment-scale water budgets (Brooks et al. 2015). Recent efforts have extended the record and detail of auxiliary data of older efforts (e.g. MOPEX) to develop longer-term streamflow and hydrometeorological forcing data for a larger number of minimally disturbed catchments, including the CONUS (Newman et al. 2015), Chile (Alvarez-Garreton et al. 2018), and Brazil (Chagas et al. 2020). The Catchment Attributes for Large-Sample Studies (CAMELS, Newman et al. (2015)) compiled high quality streamflow measurement in 671 unimpaired catchments across the CONUS, as well as climate forcing datasets (e.g. daily precipitation and temperature) and physiographic properties (e.g. land cover, topography, etc., Addor et al. (2017). CAMELS has seen widespread adoption by the hydrological community as a benchmarking tool for hydrological models (Melsen et al. 2018, Mizukami et al. 2019, Pool et al. 2019, Kratzert et al. 2023), in the development of hydrological signatures and new information theory-based approaches, and the application of novel machine learning tools (Kratzert et al. 2019). The Global River Water Quality archive (GRQA), which includes the GLObal RIver Chemistry Database (GLORICH), offers opportunities for water quality analyses across time and land scale (Hartmann et al. 2014, Virro et al. 2021). The combination of catchment attributes and matching datasets on stream water chemistry has recently been developed for Germany (Ebeling et al. 2022), however, for the CONUS this approach has not seen as much development (Arora et al. 2023). Furthermore, atmospheric deposition data is available for CONUS but has seen less inclusion in such data sets, despite the significant impact of atmospheric contribution to stream chemistry (Shao et al. 2020).

Many important insights across Earth science disciplines over the last several decades developed from investigations that combined several datasets such as long-term stream chemistry data, stream discharge, hydroclimatology, and catchment properties (e.g. vegetation, geology, topography). For example, global analyses of $CO_2$ evasion from headwater streams have helped to quantify global fluxes (Lauerwald et al. , Gaillardet et al. 1999, Raymond et al. 2013, Horgby et al. 2019). Changes in dissolved organic carbon (DOC) in stream water were partially related to changes in atmospheric deposition and acidity only viewable with longer records (Monteith et al. 2007). Stream flow and chemistry data, in particular paired concentration and discharge data, have also been shown to illuminate subsurface flow paths and chemical vertical stratification (Zhi and Li 2020, Zhi et al. 2020, Stewart et al. 2022). Applying an integrative dataset in the Northeastern United States, recent studies showed differential sensitivity of headwater catchments to reductions in $SO_4$ and $NO_3$ and resulting variations in stream DOC efflux (Adler et al. 2021, Ruckhaus et al. 2023). Importantly, this work confirmed that much of the long-term recovery from acid rain is mediated by catchment-scale processes in variable soils and bedrock as well as variable hydrological and climatic forcings. Only by aggregating data across many catchments could these interacting factors and their effect on stream chemistry be determined (Clow and Mast 2010, Harpold et al. 2010). Further, long-term water chemistry datasets have also given insight into rock weathering and solute flux estimates. For example, Godsey et al. (2009) used the GLORICH dataset that focuses on large, human impacted riverine systems and contains corresponding catchment properties and streamflow data and showed the ubiquity of 'chemostasis', i.e. solute fluxes being primarily driven by stream discharge and not variations in concentrations in many catchments. More recent work on >2,000 GLORICH catchments began to illustrate the role of aridity and catchment properties in controlling the concentration-discharge (C-Q) relationship in key solutes (Godsey et al. 2019). However, one of the limitations of current datasets like GLORICH is the lack of observed instantaneous streamflow discharge measurements and deposition chemistry, as well as important catchment properties taken from reliable sources.

There is a need for more open-source datasets that integrate information scattered across different databases and formats to present comprehensive and verified data on water chemistry, event-scale hydrology, atmospheric deposition and critical zone attributes to facilitate new ecohydrological research and understanding at broad scales. For example, Vlah et al. (2023), have combined data from watershed studies and supplemented these with watershed attributes to provide a finer spatial resolution (smaller watersheds) and finer temporal resolution. Our contribution is the compilation and release of multiple harmonized datasets that take advantage of one of the most comprehensive collections of catchment attributes across the CONUS i.e., CAMELS (Newman et al. 2015, Addor

et al. 2017). Building on CAMELS, we developed "CAMELS-Chem", a relational database that provides fast query processing, enforces data integrity, provides detailed information about current data and schema (i.e., relational structure). The database comprises individual water chemistry measurements and paired streamflow data harvested from the U.S. Geological Survey (USGS) National Water Information System (NWIS), for up to 516 catchments and 18 constituents from earliest available sample times through 2018 (USGS 2023) and earliest atmospheric deposition data from the National Atmospheric Deposition Program (NADP, available since 1985). In order to facilitate the use of the CAMELS-Chem dataset for interdisciplinary research, we explore the potential application of CAMELS-Chem to examine the interconnections among water chemistry, hydrology, atmospheric deposition, and biogeochemistry. Our investigation is guided by two main groups of questions related to data availability and novel applications: 1. How consistent is water chemistry sampling across regions, how well does sampling capture the range of discharge variability, and how coincident are various water chemistry sampling programs across the CAMELS watersheds? 2. Can we use CAMELS-Chem to explore trends in stream water and deposition chemistry, biogeochemical processes, or develop hypotheses not apparent from current process understanding?

We first introduce the methods used to develop the CAMELS-Chem database and dataset to highlight the value added by the dataset (Section 2) and then evaluate questions on data availability to illustrate the utility of the dataset for a variety of intended purposes (Section 3). We then tackle considerations of novel applications, offering examples of interdisciplinary uses that demand multiple data streams in (bio)geochemistry, ecology, and hydrology (Section 4). These examples can help illuminate how the dataset can be used and facilitate the integration of ideas, as it is often challenging to see connections across disciplinary boundaries. Many of these examples highlight the cross-disciplinary publications that have already used this dataset (Zhi et al. 2019, Zhi and Li 2020, Adler et al. 2021, Zhi et al. 2021, Li et al. 2022).

## 2 Materials and Methods

In our CONUS study area, we leveraged the CAMELS data set (Newman et al. 2015, Addor et al. 2017) compiled for 671 catchments of the US Geological Survey (USGS) National Water Information System (NWIS). These are minimally-disturbed catchments filtered from the Hydro-Climatic Data Network (Lins 2012) that have been used in previous studies (Godsey et al. 2019). Catchment attributes from this data set span categories of topography, land cover characteristics, soil characteristics, and geological characteristics, as well as climatic indices and hydrological signatures. The hydrological signatures were computed using daily discharge time series (sourced from NWIS) to calculate average values and ratios (e.g., mean daily discharge, runoff ratio, frequency of high

flows) over a multi-year period of observation (Newman et al. 2015, Addor et al. 2017); these indices describe the hydrological character of each catchment at seasonal to annual scales. To facilitate more temporally-refined ecohydrological analyses at catchment to CONUS scales, we built a relational database linking these CAMELS attributes to instantaneous discharge and streamwater chemistry time series data, as well as wet deposition data.

## 2.1 Data sources and description

From USGS National NWIS, we harvested: (1) (bio)geo-chemical stream water quality data and (2) paired instantaneous and daily mean discharge data; we also compiled (3) wet deposition data from the National Atmospheric Deposition Program. Analyses and summary figures displayed in this work cover the time period from 1980 to 2018, to be consistent with the start date of hydrological signatures computed in Newman et al. (2015). Data for the full available length of record (in some cases dating back to 1906) up through 2018 are contained in supplementary materials and in the linked data repositories. Instantaneous (15-minute to hourly) discharge data, daily mean discharge data, and (bio)geo-chemical sample data were harvested from the NWIS repository. USGS protocols provide for a consistent methodology, location, and paired discharge record for collected water quality samples. Approximately 93% of the samples in CAMELS-Chem have a paired daily discharge value; however, the coverage for paired instantaneous discharge values was lower (~15%) because not all gauges provided hourly observations or only from a more recent part of the record.

Wet deposition data were obtained NADP for the earliest availability, starting in 1985, through 2018. This data product is generally an inverse distance weighted interpolation of the wet deposition observation stations and is reported as a concentration and deposition (mass per area) in a raster data set of 2.5 km resolution; more details are provided on their website (Accessed July 1, 2023: https://nadp.slh.wisc.edu/maps-data/ntn-gradient-maps/). We compiled annual deposition data for 10 species ($H^+$, $NH_4$, $NO_3$, $NH_4$, $Cl$, $SO_4$, $Ca$, $K$, $Mg$, and $Na$) for each CAMELS-Chem watershed, relying on catchment shape files available from CAMELS. Quantum GIS 3.12 was used to calculate mean, minimum, maximum, and standard deviation of the concentration and deposition values for each species by catchment and by year.

## 2.2 Automated data acquisition and integration

Acquiring and integrating data from multiple sources can be a process replete with challenges including missing data, mis-matched sample times, inconsistent parameter names, or varying units of measure (Sprague et al. 2017, Niu et al. 2018). Fortunately, NWIS sources already have high quality records that have been quality-assured and

normalized; issues of missing data were relatively minor and limited to missing time stamps for approximately 10% of the water quality records. Additionally, use of different constituent names and analytical methods across catchments presented some issues for data harmonization and integration. To address these challenges, we used state-of-the-art methods to automate data acquisition and integration, coupled with a relational database (Bansal and Kagemann 2015). The Extract, Transform, and Load (ETL; Figure S1 in supplementary materials) framework from Pentaho Data Integration was employed to: (1) extract discharge and water quality data from siloed systems; (2) transform the data into formats to fit operational needs; and (3) load the data into a relational database to provide maximum flexibility for retrieval, exploration and analysis.

Our ETL process started with the extraction of data from USGS flat files for each gauge. These flat files comprised (bio)geochemical data, daily discharge, and instantaneous discharge data. The raw data were then loaded into a staging table for initial processing. The transformation phase involved normalizing differing units of measure across constituents (Table 1), adjusting sample times across time zones, and using sophisticated algorithms to fill missing time stamps for a subset of water quality records (next section). Large-scale corrections and data harmonization were handled using automated processes, significantly reducing the potential for human error and enhancing data consistency. Of the original 671 CAMELS watersheds, 516 watersheds had water quality data, 506 watersheds returned instantaneous or daily mean discharge data during periods of water quality sampling, and 488 returned both discharge and water quality data for the time between 1980 and 2018. For the final comprehensive data repository, we selected Oracle's free and open-source database (PostgreSQL 2020). Such a database makes data provisioning easier and is optimized to prevent data anomalies by only storing data in one place and using keys to relate different tables and data to each other.

**2.3 Data imputing, harmonizing, and harnessing**

While several packages are available to facilitate the retrieval of streamflow and water chemistry data from NWIS (e.g., "dataRetrieval" package for R), we selected the ETL framework and relational database for the advantages of this architecture in imputing, harmonizing and harnessing data.

- **Harnessing data.** Multiple decades of discharge and water chemistry data exist for the 516 catchments, comprising well over 2.67 million records. This volume of data makes it impractical to work with flat file formats that would be generated using standard data retrieval packages, particularly for broad-scale analyses across multiple catchments. A relational database provides fast query processing, enforces data integrity, provides detailed

information about current data and schema (i.e., relational structure), and represents a flexible platform to export data in a consistent format for external analysis.

- **Missing time stamps.** We applied algorithms to fill in missing time stamps for water chemistry data. We made the assumption that field technicians generally collect samples for multiple solutes at the same time, thus the recorded timestamp for one solute can be applied to other solutes taken on that day. By assuming one sampling time every day per site we obtained 397 more instantaneous discharge values (out of a total of 8,975 records without timestamps). The updated time stamps were used to fill hourly, instantaneous discharge values for roughly 30% of samples.

- **Missing discharge values.** For a small percentage (9.9%) of water chemistry records with a missing value for instantaneous discharge, the (bio)geo-chemical sample time was adjusted forward/backward to pair with the closest quarter -hour (or sometimes hourly) instantaneous discharge observation using date math programming in PostgreSQL.

- **Harmonizing data**. Through transformation steps, and before data were uploaded to the repository, we ensured consistent units of measure and normalized sample times across time zones. Also, data with somewhat different constituent names and analytical methods across catchments were aggregated (i.e, $NO_3$, $Cl$, $SO_4$) following USGS guidance (Oelsner et al. 2017).

## 2.4 Statistical Summaries and Example Analyses

Summary statistics were calculated using the database for key parameters such as flow duration curve (FDC), standard deviation, low/medium/high flow conditions, mean daily and annual flow. The FDC was calculated following the methods of Newman et al. (2015) to use all daily value and compute percent exceedance. We then report FDC information in terciles of high, normal, and low flow.

We developed two examples in Section 4 using CAMELS-Chem datasets and simple analyses. In Section 4.1 we investigate trends in atmospheric $SO_4$ deposition and resulting trends in stream water chemistry using a Mann-Kendall test for three timeframes (Hirsch and Slack 1984): 1985-2002, 2003-2012, 2003-2010. In section 4.2, we investigate weathering by developing molar ratios between $HCO_3$ to Na and Mg to Na using only low flow discharge when FDC >66%.

## 3 Results

## 3.1 Stream Water Chemistry

The CAMELS-Chem dataset comprises 18 water chemistry and property values (Table 2) and is summarized for
general water quality and physical parameters (discharge, dissolved oxygen, pH, and temperature), carbon and
nutrient species ((DOC), total organic carbon (TOC), dissolved organic nitrogen (DON), total organic nitrogen
(TON), total dissolved nitrogen (TDN), nitrate ($NO_3$)), anions (Cl, bicarbonate ($HCO_3$), sulfate ($SO_4$) and cations
(Ca, K, Mg, Na, Si, and Al; see Figure 1). While the lengths of discharge and climate records extend up to or greater
than 30 years for many CAMELS-Chem catchments, the water chemistry data are not as continuous and spatially
consistent (see Section 3.2). The total number of stream water samples varies substantially depending on the
variable of interest, which should be taken into consideration when using this dataset. For example, the dataset has
19,784 total Si measurements from 325 catchments, compared to only 10,322 DOC measurements from 179
catchments (Figure 1r and 1e, respectively).

Patterns of different water chemistry constituents reflect the broad range of geology, climate, land use history, land
cover and other factors (Figure 2). For example, water temperature shows clear latitudinal patterns in both mean
and coefficient of variation (CV, Figure 2d) that likely reflect warmer climate and greater solar radiation in southern
locations. Conversely, variables like Cl and Na have much smaller variability in mean and CV and less geographic
patterning (Figure 2b and c, respectively). Some of the biologically derived solutes such as DOC and TOC show
high variability in mean and CV across CONUS, without clear geographic patterns (Figure 2e and f). In contrast,
many of the nitrogen constituents show hot spots of agriculture in the Midwest and upper great plains (Figure 2g-
j). Similarly, ions associated with agriculture and human impacts, like Ca, K, and Mg had higher mean values and
larger CV in the central part of the CONUS (Figure 2n, k and p); whereas the geographical pattern of mean and
variance of Si and Al is less distinct (Figure 2r and s). These results demonstrate the complexity of spatial patterns
across the CONUS. In the following sections we offer more background on the strengths and weaknesses of the
dataset for different applications.

CAMELS-Chem offers long-term records for trend analysis and broad geographic coverage in catchments (Figure
1, see Figure S5 for the entire length of record). Because USGS sampling foci varied between decades, temporal
biases in the sampling record exist (Shanley et al. 2015). For example, many of the stream water constituents
impacted by acid rain (i.e., $SO_4$ and Ca) were sampled less frequently starting in the late 1990's (Fig 3m and n). In
contrast, sampling frequency for many solutes related to local water quality issues (i.e. $NO_3$, and K) have increased
in recent decades (Figure 3j and o). The $NO_3$ data are more abundant in the Midwest and along the east coast where

sampling for nutrients is common. In contrast, DOC observations highlight the location of long-term sites focused on minimally human-impacted catchments (i.e., USGS Hydrologic Benchmark Network) that have the most comprehensive sampling of all solutes (Figure 3e).

Longer records and sampling across seasons and lower streamflows increase the capability for trend detection. DO, pH, and temperature are more consistently measured (up to 516 out of the 671 CAMELS watersheds, Figure 3b-d). For example, temperature was measured 3,000-7,000 times each year, resulting in a dataset that represents >400 sites with >20 years of records and >100 samples (Figure 3d). However, temperature is measurement relatively in the field, whereas other water quality constituents require additional work, such as filtration in a laboratory. Among

these samples, weathering-related, lithogenic solutes such as Ca, K, Mg, Na and Si (Figure 3n-r) are more comprehensively sampled (i.e. longer records and more catchments) than biologically driven constituents such as DOC, TOC or DON (Figure 3e-h). For example, lithogenic solutes have around 50 sites with >20 year records and >100 samples, with a total of between 50-150 catchments and 1,000-2,000 samples each year. In contrast, biologically related solutes are sampled around 500-1,000 times per year, have <50 records that are >20 years and

>100 samples. NO3 is sampled 1,000-2,000 times per year, but much of the additional sampling is concentrated in agricultural catchments in the midwestern part of the US. Thus, data users should be aware of the different sampling frequencies at each location. Interestingly, though sampling frequency varies greatly by solute and water year, seasonal coverage is relatively even in the dataset (Figure 4, see Figure S5 for the entire length of record). Across all watersheds, CAMELS-Chem covers a wide range of hydroclimate, which offers ample opportunities for

investigating the connection between climate, catchment attributes, and stream water chemistry. We illustrate this by showing the range of some variables from the CAMELS database (Figure 5) for all 671 catchments versus the catchments sampled by CAMELS-Chem for Cl (high frequently sampled) and Al (less often sampled). The range of hydrological and meteorological conditions catchments sampled is nearly identical between by CAMELS and CAMEL-Chem catchments, is nearly identical both of which span a wide range of hydrological and

hydrometerological conditions.

## 3.3 Consistency of Sampling Across Discharge Records and Variable Hydroclimate

Because CAMELS-Chem is paired with measured discharge data, we can effectively assess and constrain the effects of discharge on water chemistry analyses. Concentration-discharge (C-Q) relationships are routinely used to compute solute loads for mass budgets (Cohn et al. 1989) and have been used to infer catchment effects on

biogeochemical cycling (Basu et al. 2010, Musolff et al. 2015). CAMELS-Chem has distinct advantages in this

context, as we used instantaneous (hourly) discharge data to supplement the NWIS database and 30-year daily discharge records are a reference for C-Q ranges. Because discharge can be quite variable at diel scales during high flow versus low flow periods, instantaneous discharge measurements are more critical on high flow days than low flow days.

To apply a C-Q analyses to a dataset, the stream chemistry sampling for the solute of interest must span a large range of discharge values. Conversely, if only low flow (baseflow) values are available, careful subsampling is required. The FDC is often used to represent variance in streamflow, and can be constructed using all daily streamflow values in ranked order. To aid assessment of the dataset in this context, we offer a visual representation of the coverage of the percent of the FDC during water chemistry sampling (i.e. highest minus lowest percentiles

of water sampling dates) for each catchment (Figure 6 and S6 for the full length of record). The CAMELS-Chem sampling covers >75% of the FDC curve in most catchments, with coverage less than 50% of the FDC in parts of the Gulf Coast and Upper Midwest areas (Figure 6). Despite the greater sampling frequency of the weathering related solutes (Figure 6m-q), all solutes show relatively high coverage of the FDC, including the biological solutes that were sampled fewer times and over shorter record lengths (e.g. Figure 6d-i). In terms of sampling consistency

and numbers across the FDC, we examined the percentage of sampling that occurred in each tercile of the FDC across solutes (Table 5), where an even sampling distribution would be 33% of samples in each tercile. There is a small bias towards high flow measurements (<33% tercile), especially for the biological solutes and many lithogenic solutes (Table 5). Over 25% of all samples are collected at low flow (>66% tercile) for all solutes (except Al) and $HCO_3$ is exclusively sampled at low and moderate flows.

**3.4 Coincidence of Sampling Across Species**

A key strength of the USGS sampling program is that a variety of water constituents are measured simultaneously, which allows concentration ratios and mixing models to be more readily developed (Godsey et al. 2019). We report this information as a table of percent of coincident samples (Table 4). For example, we see that daily discharge is co-sampled with the water chemistry constituents >90% of the time (right most column), however only about 10-

30% of discharge sampling dates have a water chemistry measurement (bottom most row). Lithogenic solutes and some anions appear to be co-sampled over 90% of the time, while many of the biological solutes were sampled less often. In this case, between 30%-80% of samples had coincident ion chemistry, with constituents like DOC and $NO_3$ being more likely to have coincident ion chemistry than TN, TON, DON, and DO. The nitrogen sampling was often coincident between TN and $NO_3$, with the other nitrogen species sampled less coincidentally.

## 3.5. Atmospheric deposition data

Wet deposition data obtained from NADP started in 1985 and ended (at the time of publication) in 2018. Atmospheric deposition needs to be considered when evaluating water chemistry patterns and especially for weathering studies, the contribution of atmospheric deposition needs to be corrected for (Berner and Berner 2012). For example, Cl and Na deposition values are higher in coastal areas (Figure 7e and j), while $NH_4$ and $NO_3$ deposition vales are higher in places where anthropogenic inputs of fertilizer are high (Figure 7b and c). Ca typically has higher values away from coastal areas and is strongly impacted by local bedrock and soil composition (Berner and Berner 2012). In many cases these patterns are consistent with patterns in stream chemistry (e.g. Figure 1j for stream $NO_3$).

## 4 Example Analyses Using CAMELS-Chem

The CAMELS-Chem combines stream water chemistry with deposition and catchment properties for i) trend attribution, ii) process understanding, and iii) generating new hypotheses of how systems work. Our goal for this section is to demonstrate the different applications of the CAMELS-Chem dataset in this context, its potential limitations, and to motivate future work. For this we highlight select examples for data use ranging from trends in $SO_4$ deposition and stream chemistry by bedrock lithology to biogeochemical processes controlling stream chemistry. We finish by offering examples on how such datasets offer opportunities for hypothesis generation using "pattern to process" frameworks.

## 4.1 Trend Detection and Attribution

Trend detection and attribution is important for assessing long-term changes from climate change, atmospheric deposition, vegetation change, or other disturbance vectors. The filtering and querying capabilities of the CAMELS-Chem database offer important advantages for large-scale studies designed to detect trends in stream water chemistry in response to disturbance at regional to continental scales across minimally disturbed catchments. The CAMELS-Chem dataset we discuss here focuses on a shorter record from 1980-2018 but contains data from before 1950 for most constituents (Figure S5). Individual sites have >100 samples over >20 years for most constituents, as well as sparser sampling at other gauges (Figure 1 and Table 4).

CAMELS-Chem provides new potential to analyze the effects of acid deposition on long-term stream chemistry trends across a range of hydrological conditions. The Industrial Revolution caused a rapid increase in fossil fuel emissions, which introduced acid anions ($SO_4$ and $NO_3$) in excess of background conditions leading to acidic

precipitation throughout many industrialized regions (Newell and Skjelkvåle 1997). The Clean Air Act in 1970 and subsequent amendment in 1990 led to major reduction in air pollution as apparent in progressively decreasing deposition amounts (Baumgardner et al. 2002, Lloret and Valiela 2016). Many (but not all) CAMELS-Chem sites have >100 $SO_4$ stream samples spanning over 20 years of record (Figure 2m). As expected, when plotting trends in $SO_4$ stream chemistry and wet deposition for an earlier timeframe (1985-1992, Figure 8a) decreasing trends in $SO_4$ deposition and corresponding decreasing trends in $SO_4$ stream chemistry are apparent. Wet deposition trends remain decreasing in the following two decades (1992-2002 and 2002-2010), but without much response in $SO_4$ stream chemistry. Our results are in agreement with previous findings of declining $SO_4$ deposition following the 1990 Clear Air Amendments (Figure 7, Garmo et al. (2014)). For example, in the Northeastern United States, $SO_4$ stream chemistry has generally responded to declines in $SO_4$ deposition (McHale et al. 2017, Siemion et al. 2018). This initial analysis provides a starting point for hypothesis testing - for example, on the role of catchment attributes such as the dominant geology on mitigating the effects of changes in atmospheric deposition (Figure 8a-c).

## 4.2 Improving Process Understanding

One of the key uses of long-term and large sample datasets is increasing and testing process understanding. Often this takes the form of hypotheses or models being applied or tested using a large dataset to test transferability and scalability. The CAMELS-Chem dataset has already been applied to several process understanding based studies with success. For example, (Li et al. 2022) developed a reactor versus transporter model for arid and humid catchments and used the CAMELS-Chem database for 12 constituents. The breadth of climate gradients and sampling across streamflows was critical to the findings of climate controls on river chemistry. In another example use of CAMELS-Chem, used DIC stream chemistry to show seasonal changes were controlled by CO2 concentration distribution with depth, while long-term DIC concentrations were controlled by climate. In these examples, availability of high-resolution discharge data improved the process inferences possible.

CAMELS-Chem datasets are particularly useful when different constituents are related to one another, such as discharge versus concentration relationships or molar ratios between different species. For example, to display the impact of major rock classes (i.e. silicates, carbonates, evaporites) on riverine composition, the use of molar ratios for geogenic species (Ca/Na, Mg/Na, $HCO_3$/Na) instead of absolute concentrations is useful, because large differences in concentrations between solid and liquid phases make comparisons difficult. Riverine composition is often used as an indicator for weathering rates and to draw inferences at larger scales. A classic example is the study by Gaillardet et al. (1999) where 60 of the world largest rivers were used to show a strong role of bedrock

lithology on weathering rates from Si effluxes. We display these molar ratios of stream water composition for all CAMELS-Chem sites as a function of bedrock lithology including igneous, metamorphic and sedimentary rocks (Figure 9, see Figure S7 for the entire length of record). The lower tercile encompasses more Ca and Mg samples than $HCO_3$ and Na samples based on overall sampling frequency (Figure 3m and o vs. Figure 3k and p, respectively). Similar to previous studies (Gaillardet et al. 1999), we observe an expected pattern based on lithology, with catchments underlain by carbonate plotting in the upper right (i.e., high Ca/Na, Mg/Na, and $HCO_3$/Na ratios) and unconsolidated sediments plotting in the lower left (e.g., low Ca/Na, Mg/Na, and $HCO_3$/Na ratios). These results are consistent with the high weathering rates of carbonates, where even small amounts of carbonate lithology lead to significant shifts to higher Ca/Na (calcite endmember) and Mg/Na (presence of dolomite) ratios. Although beyond the scope of this work, CAMELS-Chem gives sufficient samples to provide uncertainty estimates in Figure 9, particularly given the uneven number of samples and distribution across solutes (Figure 1). Including baseflow index further reveals higher baseflow in carbonates-underlain catchments (Figure 9), which is consistent with fractures and highly conductive conduits that are common in carbonate aquifers (Hartmann et al. 2009). In contrast, unconsolidated sediments tend to have low weathering rates and low baseflow index (Figure 9).

## 4.3 Hypotheses Generation

Large sample datasets are necessary for most data-driven approaches that can point us towards new controls and interactions, but domain knowledge is required to ascribe meaning (Goldstein et al., 2018) and refine hypotheses for further testing using process-based methods. Many new machine learning (ML) and artificial intelligence (AI) techniques are capable of determining potential linkages that are not apparent with conventional frameworks and statistical tools (Reichstein et al., 2019). Most of these new ML/AI techniques require large co-sampled datasets across a range of environmental and state conditions. For example, Underwood et al. (2023) used DOC concentrations from CAMELS-Chem paired with an Evolutionary Algorithm (EA) to develop new hypothesis of the controls on large-scale patterns. EA has several advantages over logistic regression. First, the EA can be applied to nonparametric data and is robust to varying data types (nominal, ordinal, continuous), skewed distributions, bounded data, censored data (e.g., water quality data that have a minimum or maximum reporting limit), and missing values (Anderson et al. 2020, Hanley et al. 2020). These types of ML/AI have distinct advantages when using water chemistry data like CAMELS-Chem, which often having missing or bounded data that is non-parametric. Future efforts using EA, or similar ML/AI techniques, could categorize different constituents (e.g., nitrogen, phosphorus, etc.) into high and low concentrations to explore controls on alternate catchment dynamics

(e.g. weathering, nutrient cycling, etc.). The same tool could be applied to data collected at high temporal resolution (e.g., nutrient time series) to suggest possible hourly to seasonal scale controls in future research. Although applications of ML/AI are nascent, datasets like CAMELS-Chem are fundamental to their application and the advancement of new hypotheses in the fields of biogeochemistry and ecohydrology.

## 5 Summary and Conclusions

We developed and released CAMELS-Chem, as a flat .csv file and a relational database comprising water chemistry measurements, corresponding instantaneous discharge, and wet deposition data. As a relational database, this provides fast query processing, enforces data integrity, provides detailed information about current data and schema (i.e., relational structure), and represents a flexible platform to export data in a consistent format for external analysis. The accompanying dataset available on Hydroshare will be sufficient for most applications and the relational database is available upon request due to its large size.

We found that the CAMELS-Chem database samples across most of the CONUS with sufficient sampling across regions and climates (Figure 5), discharge variability (Table 5), and coincident across a wide variety of constituents (Table 4). One of the key requirements for most studies are long-term water chemistry datasets with regular sampling. We show that sampling of different constituents varies spatially and temporally (Figure 1), reflecting changing priorities and budgets within the USGS. However, records beginning in 1980 (or earlier) span most of the CONUS for most constituents (Figure 1) and reflect semi-regular sampling that was similar across seasons (Figure 4). However, the user needs to take limitations of these data into account, such as, some constituents (i.e. water temperature, cations, etc.) are more regularly sampled than others (i.e. DOC, Al, etc.). Using FDC we show that water quality sampling spans discharge variability sufficiently (Table 3), making it suitable for constructing long-term concentration-discharge relationships and producing flow-weighted load estimates. Finally, we show that coincident sampling of water chemistry by the USGS, with variables like pH and water temperature is necessary for many hypotheses testing or modelling, such as developing molar ratios or training process-based geochemical reaction models.

CAMELS-Chem offers unique aspects for trend detection and attribution, by including long-term atmospheric deposition data and consistent daily climate data. We show that CAMELS-Chem allows for detection of long-term changes in $SO_4$ in stream water due to the Clean Air Act and consistent with other studies. Indeed, CAMELS-Chem has already shown its utility is by testing and improving process understanding. For example, we show that the coincident sampling of discharge helps to improve understanding of climate controls on DIC specifically

(Stewart et al. 2022) and a large variety of nutrient and geogenic constituents (Li et al. 2022). Another example shows the utility of coincident sampling for developing molar ratios and improving understanding of weathering processes (Figure 7). Finally, we show how data-driven ML/AI approaches could help generate new hypotheses and expose linkages not evident with current process understanding. The example by Underwood et al. (2023), shows how ML/AI techniques can be applied to CAMELS-Chem to elucidate new hypotheses for the continental-scale controls on DOC. New applications of ML/AI have the potential to better take advantage of water chemistry data that have issues of missing data, bounded data, and are nonparametric. Despite limitations in sampling frequency and record length for some constituents, CAMELS-Chem offers a unique 'off-the-shelf' stream water chemistry and wet deposition dataset across catchments with varying climate and physiographic properties.

## Code and Data Availability

The dataset is available on Hydroshare link https://www.hydroshare.org/resource/841f5e85085c423f889ac809c1bed4ac/. Code used will be made available GitHub and the SQL database is available from University of Vermont co-authors upon request.

## Author Contributions

G.S., A.H., and J.P. conceived and planned the study. G.S., T.A., D.K., H.M. and K.U. contributed to the analysis. L.L, J.P., T.A., D.R. contributed to the case studies, G.S. and I.U. curated the database. All authors contributed to the writing of the paper.

## Competing Interests

The authors declare no competing interests. We acknowledge funding from NSF EAR grants 1724171 and 2012123.

**Table 1:** NADP depositional dataset, attribute, name, concentration and deposition units.

| Attribute | Name | Concentration units | Deposition units |
|---|---|---|---|
| $H^+$ | Hydrogen | mg/l | kg/ha |
| $NH_4$ | Ammonium | mg/l | kg/ha |
| $NO_3$ | Nitrate | mg/l | kg/ha |
| $NO_3 + NH_4$ | Inorganic Nitrogen | mg/l N | kg/ha |
| Cl | Chloride | ug/l | kg/ha |
| $SO_4$ | Sulfate | mg/l | kg/ha |
| Ca | Calcium | mg/l | kg/ha |
| K | Potassium | mg/l | kg/ha |
| Mg | Magnesium | mg/l | kg/ha |
| Na | Sodium | mg/l | kg/ha |

**Table 2:** The stream water chemistry datasets in the CAMELS-Chem dataset including attribute, name, abbreviation in the database, description, units and USGS parameter codes.

| Attribute | Name | Database abbreviation | Description | Units | USGS parameter code(s) |
|---|---|---|---|---|---|
| Q | discharge | q | Stream discharge, 61 is instantaneous | cfs | 00060, 00061 |
| DO | Dissolved Oxygen | o | Water, unfiltered | mg/l | 00300 |
| pH | pH | ph | Water, unfiltered, field | std units | 00400 |
| pH_l | pH | ph2 | Water, unfiltered, laboratory | std units | 00403 |
| temp | water temperature | temp | Water, field | degree C | 00010 |
| DIC | Dissolved Inorganic Carbon | DIC | Water, unfiltered | mg/l | 00691 |
| DOC | Dissolved Organic Carbon | doc | Water, filtered | mg/l | 00681 |
| TOC | Total Organic Carbon | toc | Water, unfiltered | mg/l | 00680 |
| DON | Dissolved Organic Nitrogen | don | Water, filtered | mg/l | 00607 |
| TON | Total Organic Nitrogen | ton | Water, unfiltered | mg/l | 00605 |
| TDN | Total Dissolved Nitrogen | tn | Water, filtered [nitrate + nitrite + ammonia + organic-N] | mg/l | 00602 |
| $NO_3$ | Nitrate | no3 | Water, filtered and total | mg/l as N | 00618, 00620 |
| Cl | Chloride | cl | Water, filtered and total | mg/l | 00940, 99220 |
| Alkalinity | Alkalinity | alk | Alkalinity, water, filtered, Gran titration, laboratory | mg/l as $CaCO_3$ | 00410, 00417, 29803 |
| $HCO_3$ | Bicarbonate | hco3 | Water, filtered, field, inflection-point (incremental titration method) | mg/l | 00453 |
| $SO_4$ | Sulfate | so4 | Water, filtered and total | mg/l | 00945, 00946, 99127 |
| Ca | Calcium | ca | Water, filtered | mg/l | 00915 |
| K | Potassium | k | Water, filtered | mg/l | 00935 |
| Mg | Magnesium | mg | Water, filtered | mg/l | 00925 |
| Na | Sodium | na | Water, filtered | mg/l | 00930 |
| Si | Silica | si | Water, filtered | mg/l | 00955 |
| Al | Aluminium | al | Water, filtered | μg/l | 01106 |

**Table 3:** Summary statistics of CAMELS-Chem. The stream water chemistry datasets in the CAMELS-Chem dataset including attribute, name, abbreviation in the database, description, units and USGS parameter codes. Statistics include the total number of gauges that measure each attribute; the median number of gauges that measure each attribute each year; the median number of measurements made each year; the range of median attribute values across all gauges; the median first year of record across

| Attribute | Total no. of gauges | Median no. of gauges per year (min., max.) | Median no. of measurements per gauge ($25^{th}$, $75^{th}$ %) | Range of median values across gauges (min., max.) | Median first year of record (min., max.) | Median final year of record (min., max.) |
|---|---|---|---|---|---|---|
| Q | 509 | 196 (94, 270) | 110 (15, 197) | (0.1, 10800) | 1980 (1980, 2018) | 2009 (1980, 2018) |
| DO | 381 | 81 (54, 140) | 27 (4, 88) | (2, 13) | 1985 (1980, 2017) | 2004 (1980, 2018) |
| pH | 420 | 110 (68, 200) | 40 (6, 119) | (4, 9) | 1981 (1980, 2017) | 2002 (1980, 2018) |
| pH_l | 329 | 72 (3, 149) | 28 (3, 76) | (4, 8) | 1983 (1980, 2015) | 1998 (1981, 2018) |
| temp | 515 | 197 (92, 294) | 111 (16, 194) | (0.5, 27) | 1980 (1980, 2018) | 2008 (1980, 2018) |
| DIC | 5 | 1 (1, 3) | 1 (1, 1) | (1, 24) | 1981 (1981, 1993) | 1981 (1981, 1993) |
| DOC | 179 | 22 (3, 60) | 7 (2, 32) | (0.3, 44) | 1993 (1980, 2017) | 1999 (1980, 2018) |
| TOC | 165 | 19 (11, 82) | 14 (4, 29) | (0.8, 48) | 1981 (1980, 2016) | 1991 (1980, 2018) |
| DON | 183 | 15 (2, 65) | 9 (2, 23) | (0.01, 2) | 1989 (1980, 2018) | 1995 (1980, 2018) |
| TON | 258 | 43 (31, 93) | 24 (4, 50) | (0.04, 8) | 1985 (1980, 2016) | 1999 (1980, 2018) |
| TDN | 178 | 14 (1, 67) | 10 (2, 21) | (0.1, 8) | 1988 (1980, 2015) | 1995 (1980, 2018) |
| $NO_3$ | 310 | 57 (35, 104) | 13 (3, 52) | (0.0, 8) | 1991 (1980, 2018) | 2002 (1980, 2018) |
| Cl | 404 | 82 (58, 160) | 15 (3, 71) | (0.1, 5500) | 1984 (1980, 2017) | 2002 (1980, 2018) |
| Alkalinity | 20 | 4 (1, 10) | 8 (3, 22) | (5, 729) | 1994 (1988, 2014) | 1996 (1988, 2014) |
| $HCO_3$ | 170 | 20 (1, 69) | 20 (3, 34) | (1, 769) | 1994 (1986, 2015) | 1997 (1988, 2018) |
| $SO_4$ | 384 | 82 (57, 166) | 18 (3, 74) | (0.09, 1200) | 1982 (1980, 2018) | 2002 (1980, 2018) |
| Ca | 360 | 73 (54, 151) | 22 (3, 77) | (0.5, 224) | 1982 (1980, 2018) | 1997 (1980, 2018) |
| K | 348 | 72 (48, 150) | 22 (3, 76) | (0.1, 27) | 1982 (1980, 2018) | 1997 (1980, 2018) |
| Mg | 360 | 73 (54, 151) | 22 (3, 77) | (0.02, 80) | 1982 (1980, 2018) | 1997 (1980, 2018) |
| Na | 352 | 72 (48, 151) | 21 (3, 76) | (0.3, 3900) | 1982 (1980, 2018) | 1997 (1980, 2018) |
| Si | 325 | 53 (32, 149) | 22 (3, 71) | (1, 90) | 1981 (1980, 2015) | 1996 (1980, 2018) |
| Al | 163 | 25 (5, 70) | 18 (4, 35) | (2, 604) | 1984 (1980, 2014) | 1996 (1980, 2018) |

all gauges; and the median final year of record across all gauges.

**Table 4:** The percentage of samples that were sampled at flows with exceedance probabilities on the flow duration curve (FDC) < 33% (higher flows) and >66% (lower flows) for water years 1980-2018. See table 1 for parameter codes.  Note: we do not show alkalinity or DIC here because coverage is so limited in the dataset (Table 2).

| Attribute | FDC <33% (%) | >66% (%) | Attribute | FDC <33% (%) | >66% (%) |
|---|---|---|---|---|---|
| DO | 35 | 34 | Cl | 40 | 31 |
| pH (field) | 37 | 33 | $HCO_3$ | 39 | 30 |
| Temperature | 41 | 29 | $SO_4$ | 41 | 30 |
| $HCO_3$ | 0 | 83 | Ca | 40 | 31 |
| DOC | 48 | 25 | K | 40 | 31 |
| TOC | 38 | 36 | Mg | 40 | 30 |
| DON | 46 | 27 | Na | 40 | 31 |
| TON | 42 | 30 | Si | 42 | 29 |
| TDN | 49 | 26 | Al | 51 | 22 |
| $NO_3$ | 43 | 28 | | | |

**Table 5:** Percent of samples stream water chemistry datasets in CAMELS-Chem for all constituents. Table is read as the percent of co-sampling with column constituent in all samples in the top right of table and the row constituent in the bottom left of table. For example, 95% of Temperature (Temp) have discharge (Q) samples and 88% of Q samples have Temp samples. See table 1 for abbreviations for USGS parameter codes for most solutes. Note: we do not show alkalinity or DIC here because coverage is so limited in the dataset (Table 2).

| | Temp | Q | DO | Cl | Na | K | Mg | Si | Al | N | TDN | NO$_3$ | SO$_4$ | Ca | DOC | TOC | pH | Alk | HCO$_3$ | DON |
|---|---|---|---|---|---|---|---|---|---|---|---|---|---|---|---|---|---|---|---|---|
| **Temp** | 100 | 95 | 37 | 31 | 28 | 28 | 29 | 26 | 9 | 14 | 7 | 18 | 30 | 29 | 13 | 7 | 28 | 1 | 6 | 6 |
| **Q** | 88 | 100 | 33 | 30 | 26 | 26 | 27 | 24 | 9 | 13 | 7 | 18 | 28 | 27 | 14 | 6 | 26 | 1 | 6 | 5 |
| **DO** | 96 | 92 | 100 | 52 | 49 | 48 | 51 | 42 | 14 | 32 | 11 | 30 | 50 | 51 | 14 | 17 | 48 | 1 | 16 | 13 |
| **Cl** | 86 | 91 | 56 | 100 | 83 | 83 | 85 | 75 | 24 | 24 | 15 | 43 | 88 | 85 | 38 | 11 | 74 | 2 | 17 | 10 |
| **Na** | 89 | 90 | 59 | 93 | 100 | 97 | 100 | 86 | 27 | 26 | 16 | 43 | 93 | 99 | 40 | 11 | 79 | 2 | 17 | 11 |
| **K** | 90 | 90 | 60 | 94 | 100 | 100 | 100 | 86 | 27 | 26 | 16 | 44 | 95 | 99 | 40 | 11 | 79 | 2 | 17 | 11 |
| **Mg** | 89 | 90 | 60 | 92 | 96 | 94 | 100 | 83 | 28 | 25 | 16 | 42 | 91 | 99 | 39 | 12 | 79 | 2 | 16 | 11 |
| **Si** | 87 | 90 | 55 | 92 | 93 | 91 | 93 | 100 | 27 | 28 | 20 | 48 | 91 | 93 | 45 | 14 | 81 | 2 | 19 | 16 |
| **Al** | 89 | 96 | 54 | 88 | 87 | 86 | 93 | 80 | 100 | 24 | 23 | 58 | 84 | 93 | 44 | 8 | 90 | 2 | 17 | 6 |
| **N** | 91 | 91 | 80 | 56 | 54 | 53 | 54 | 53 | 16 | 100 | 25 | 36 | 54 | 54 | 13 | 27 | 52 | 1 | 15 | 33 |
| **TDN** | 93 | 96 | 56 | 69 | 67 | 67 | 67 | 77 | 29 | 50 | 100 | 70 | 68 | 67 | 52 | 28 | 69 | 0 | 14 | 54 |
| **NO$_3$** | 88 | 95 | 58 | 76 | 67 | 67 | 68 | 69 | 28 | 27 | 26 | 100 | 74 | 68 | 50 | 12 | 65 | 1 | 21 | 16 |
| **SO$_4$** | 90 | 90 | 58 | 95 | 90 | 89 | 91 | 81 | 25 | 25 | 16 | 46 | 100 | 90 | 41 | 11 | 77 | 2 | 18 | 11 |
| **Ca** | 89 | 90 | 60 | 92 | 96 | 94 | 100 | 83 | 28 | 25 | 16 | 42 | 91 | 100 | 39 | 12 | 79 | 2 | 16 | 11 |
| **DOC** | 85 | 98 | 35 | 89 | 82 | 82 | 83 | 87 | 29 | 13 | 26 | 66 | 89 | 83 | 100 | 7 | 73 | 1 | 17 | 14 |
| **TOC** | 93 | 81 | 82 | 47 | 44 | 43 | 47 | 51 | 10 | 51 | 26 | 31 | 43 | 47 | 13 | 100 | 61 | 0 | 9 | 26 |
| **pH** | 90 | 91 | 59 | 85 | 81 | 79 | 84 | 77 | 29 | 26 | 17 | 43 | 82 | 83 | 36 | 16 | 100 | 2 | 16 | 10 |
| **Alk** | 97 | 98 | 96 | 96 | 92 | 91 | 94 | 92 | 31 | 29 | 6 | 53 | 92 | 94 | 39 | 1 | 94 | 100 | 97 | 2 |
| **HCO$_3$** | 93 | 96 | 93 | 87 | 78 | 78 | 78 | 80 | 25 | 34 | 16 | 63 | 86 | 78 | 38 | 10 | 72 | 8 | 100 | 14 |
| **DON** | 91 | 95 | 79 | 58 | 57 | 57 | 57 | 74 | 9 | 80 | 67 | 51 | 56 | 57 | 35 | 35 | 52 | 0 | 15 | 100 |

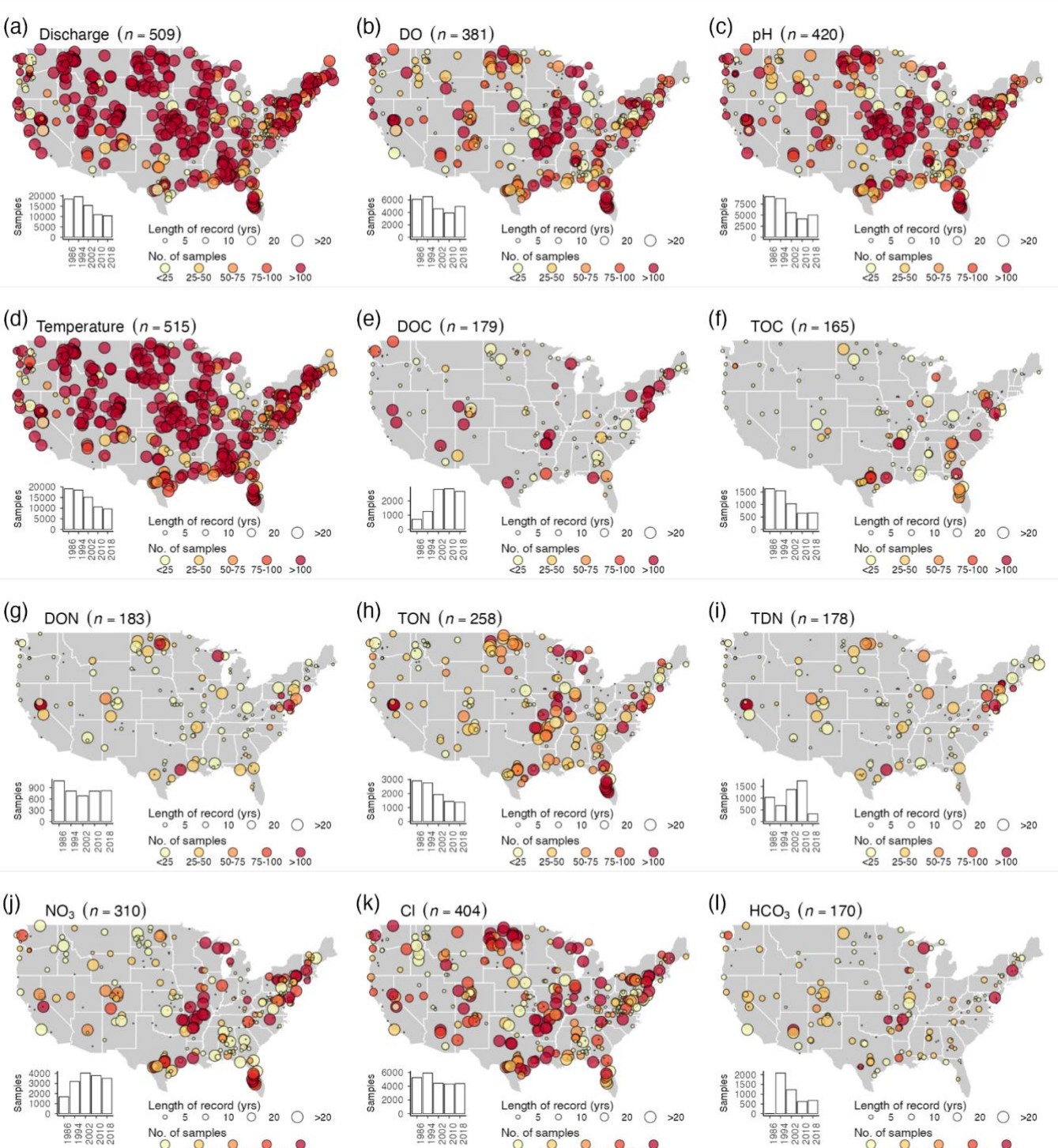

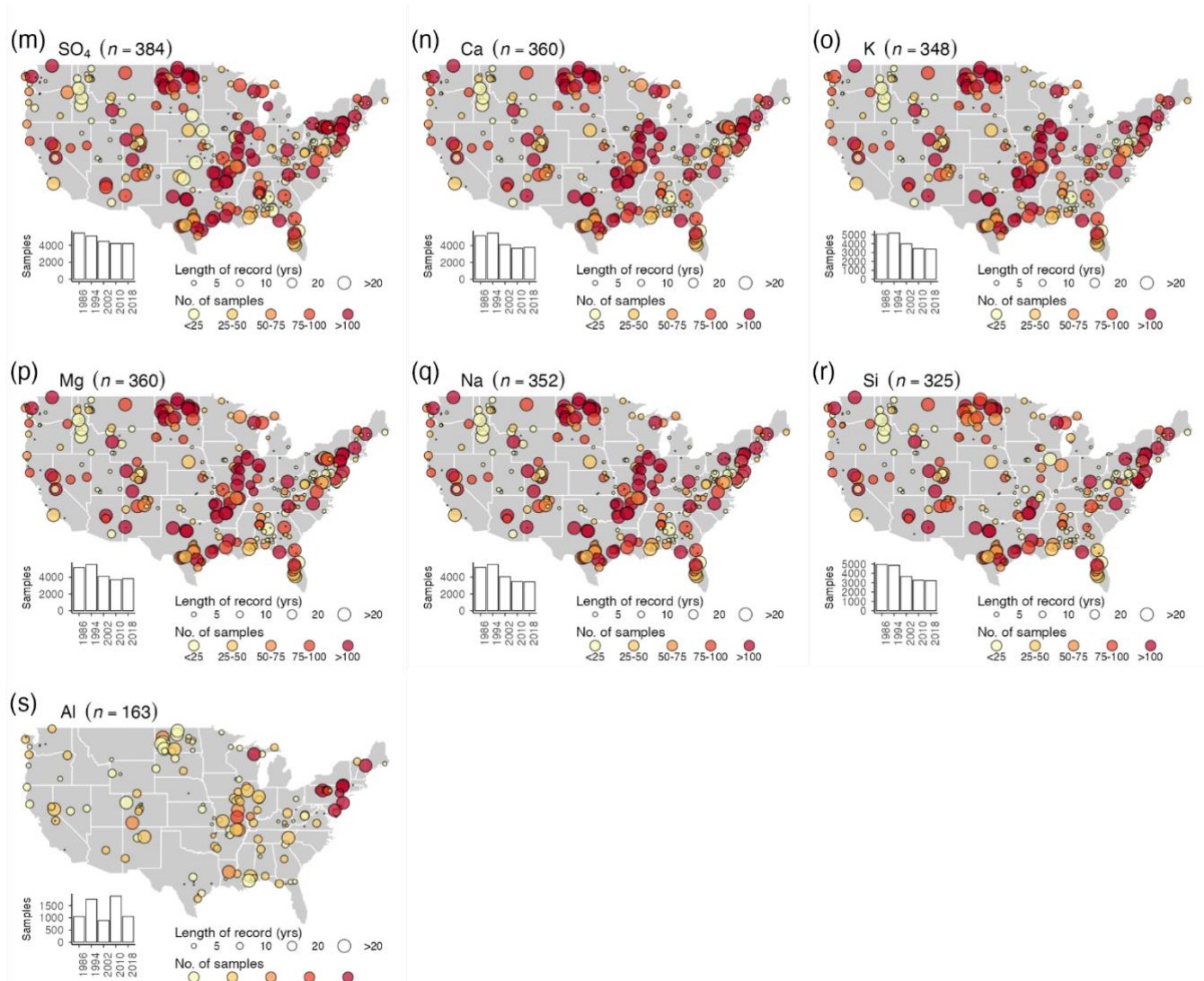

**Figure 1:** The number of samples (symbol color) and length or record (symbol size) for the stream data between 1980 and 2018 for all flow conditions (a) discharge, (b) dissolved oxygen (DO), (c) pH (field), (d) temperature, (e) dissolved organic carbon (DOC), (f) total organic carbon (TOC), (g) dissolved organic nitrogen (DON), (h) total organic nitrogen (TON), (i) total dissolved nitrogen (TDN), (j) nitrate ($NO_3$), (k) Cl, (l) bicarbonate ($HCO_3$), (m) sulfate ($SO_4$), (n) Ca, (o) K, (p) Mg, (q) Na, (r) Si, and (s) Al. The inset histogram shows the number of samples by 8-year periods.

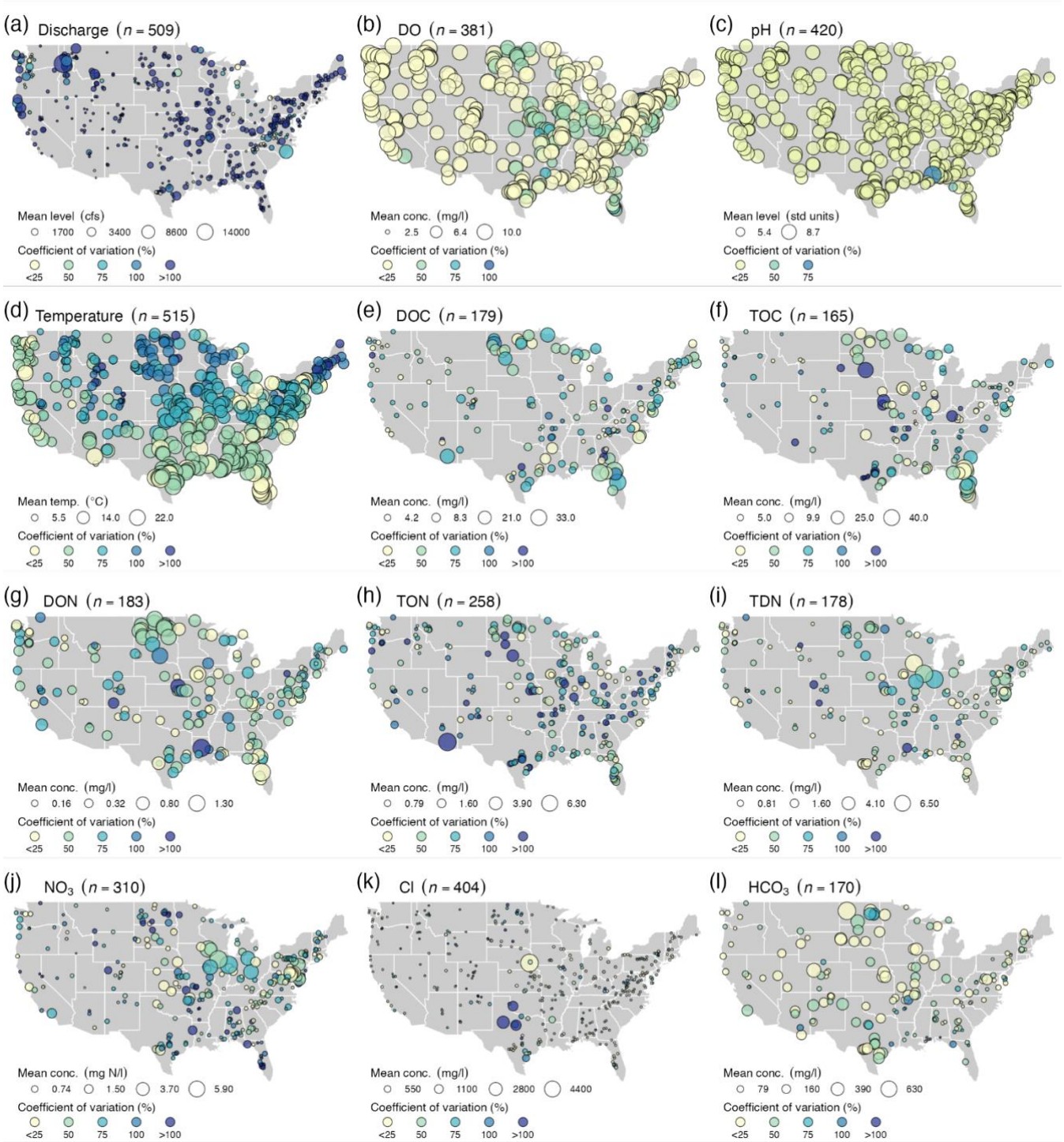

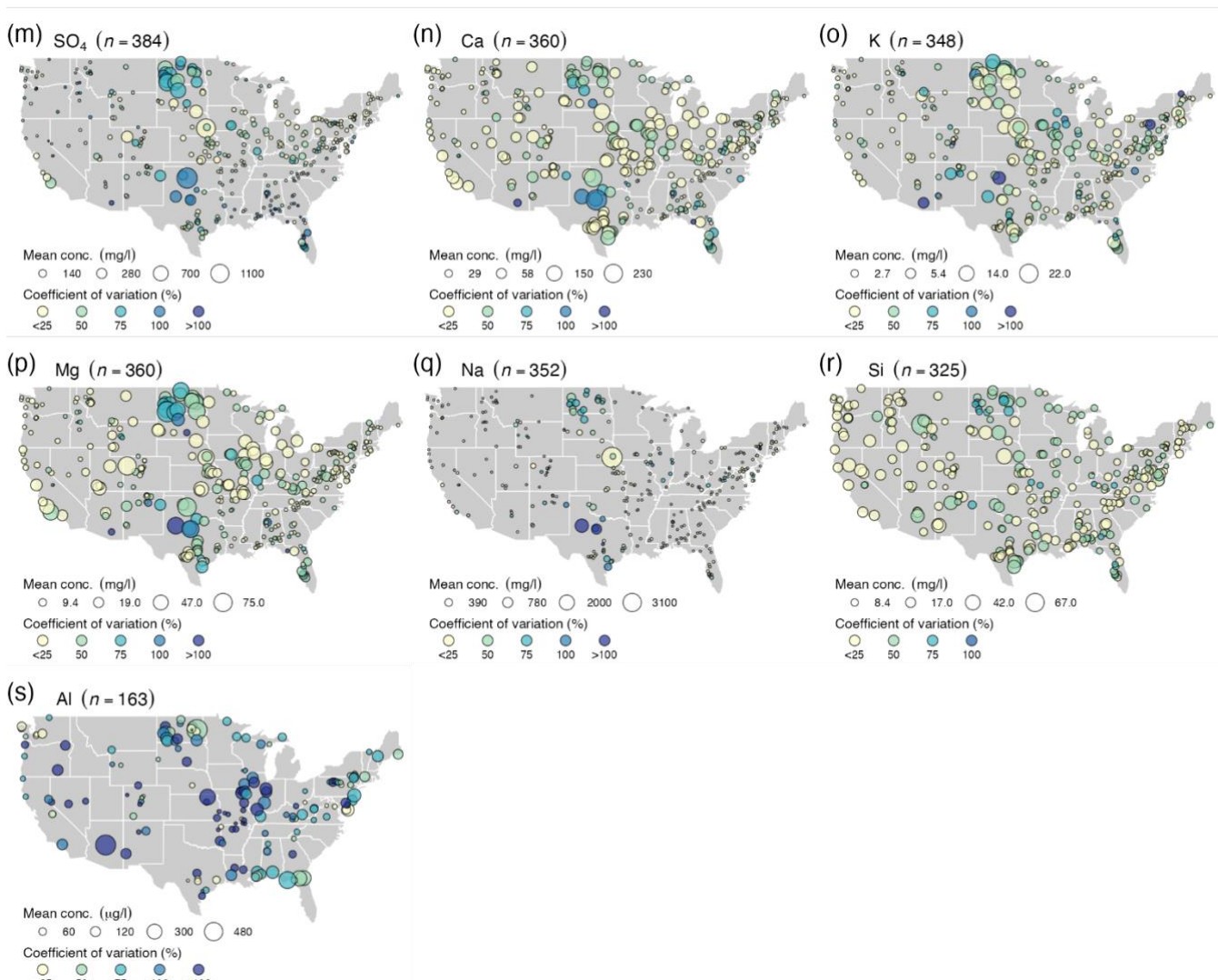

**Figure 2:** The mean concentration (symbol size) and coefficient of variation (in %, symbol color) for the stream data between 1980 and 2018 for (a) discharge, (b) dissolved oxygen (DO), (c) pH (field), (d) temperature, (e) dissolved organic carbon (DOC), (f) total organic carbon (TOC), (g) dissolved organic nitrogen (DON), (h) total organic nitrogen (TON), (i) total dissolved nitrogen (TDN), (j) nitrate ($NO_3$), (k) Cl, (l) bicarbonate ($HCO_3$), (m) sulfate ($SO_4$), (n) Ca, (o) K, (p) Mg, (q) Na, (r) Si, and (s) Al. Note: we do not show alkalinity or DIC here because coverage is so limited in the dataset (Table 2).

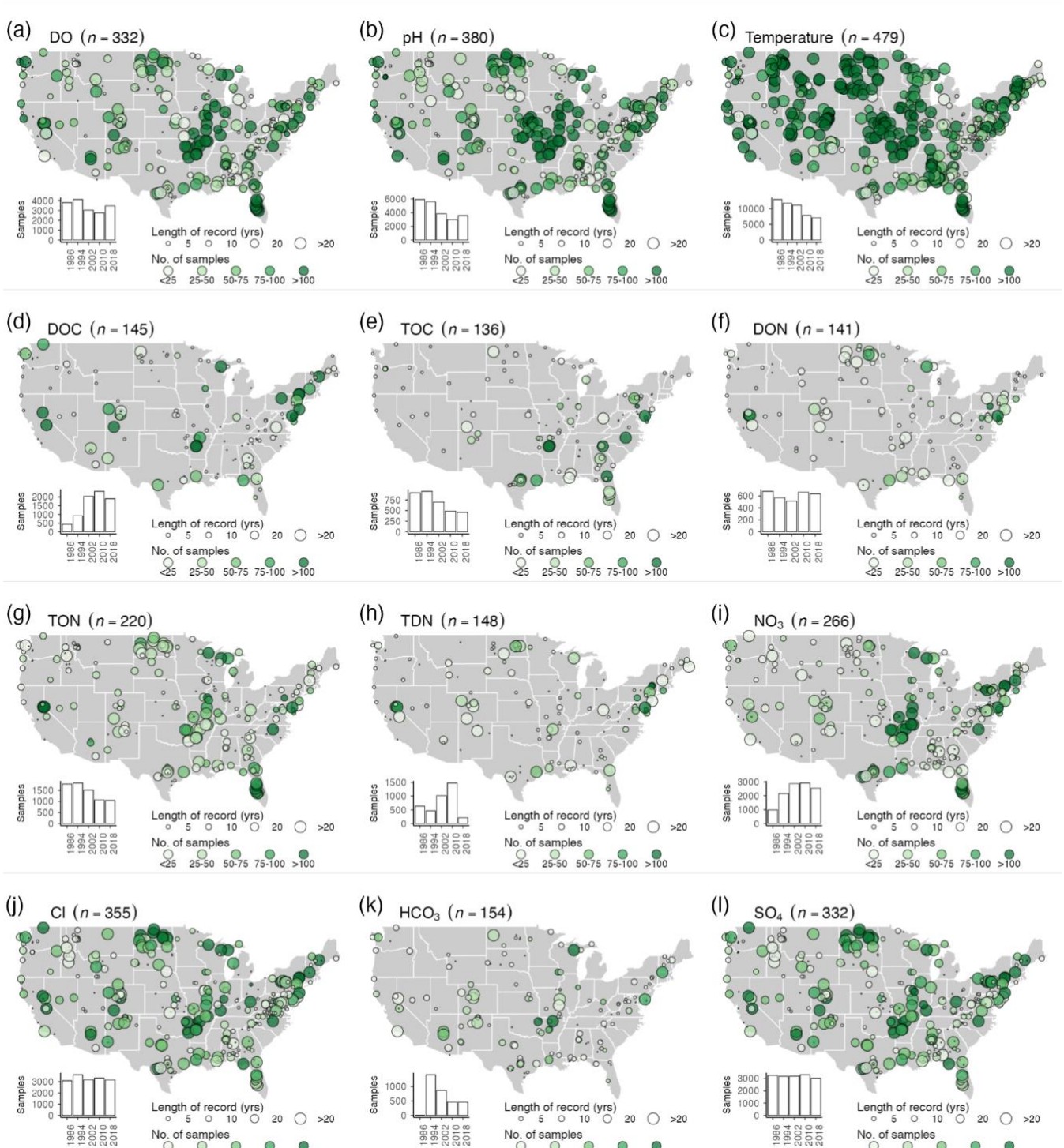

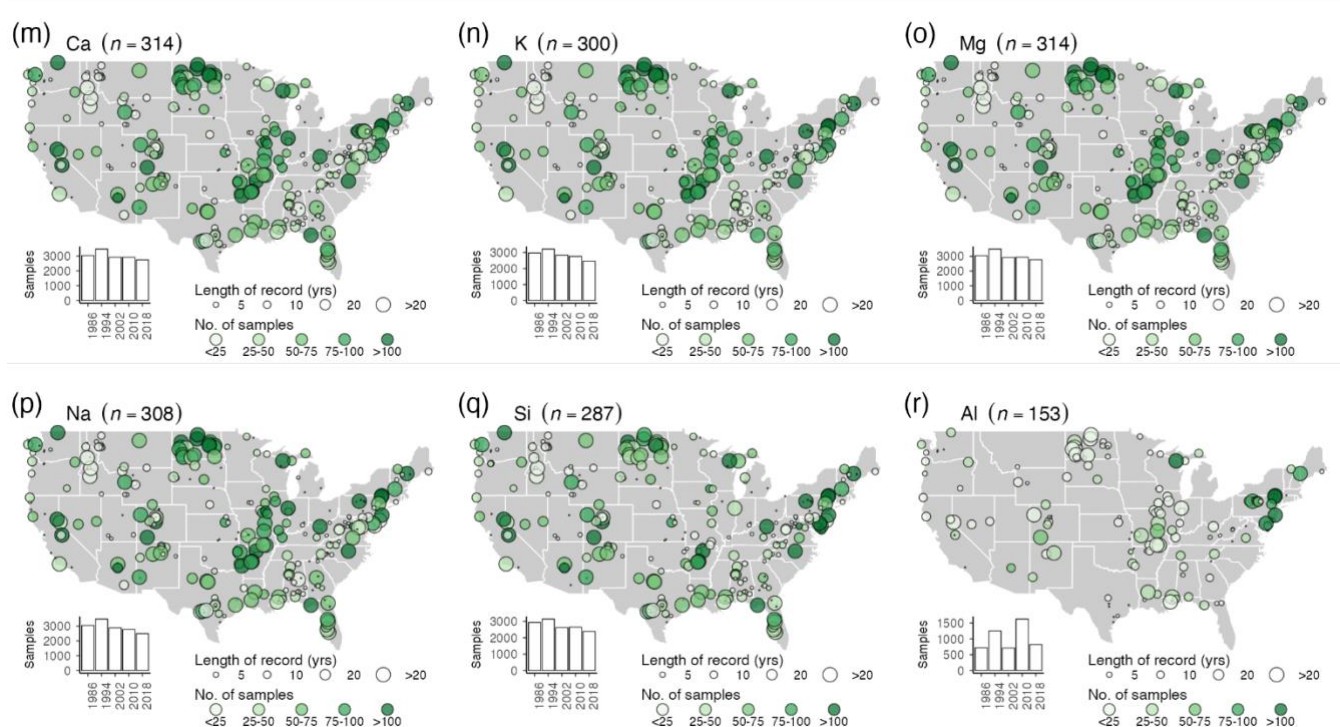

**Figure 3:** The number of samples (symbol color) and length or record (symbol size) for the stream data at low flows (flow duration curve >66%) between 1980 and 2018 for (a) discharge, (b) dissolved oxygen (DO), (c) pH (field), (d) temperature, (e) dissolved organic carbon (DOC), (f) total organic carbon (TOC), (g) dissolved organic nitrogen (DON), (h) total organic nitrogen (TON), (i) total dissolved nitrogen (TDN), (j) nitrate ($NO_3$), (k) Cl, (l) bicarbonate ($HCO_3$), (m) sulfate ($SO_4$), (n) Ca, (o) K, (p) Mg, (q) Na, (r) Si, and (s) Al. The inset histogram shows the number of samples by 8-year periods. Note: we do not show alkalinity or DIC here because coverage is so limited in the dataset (Table 2).

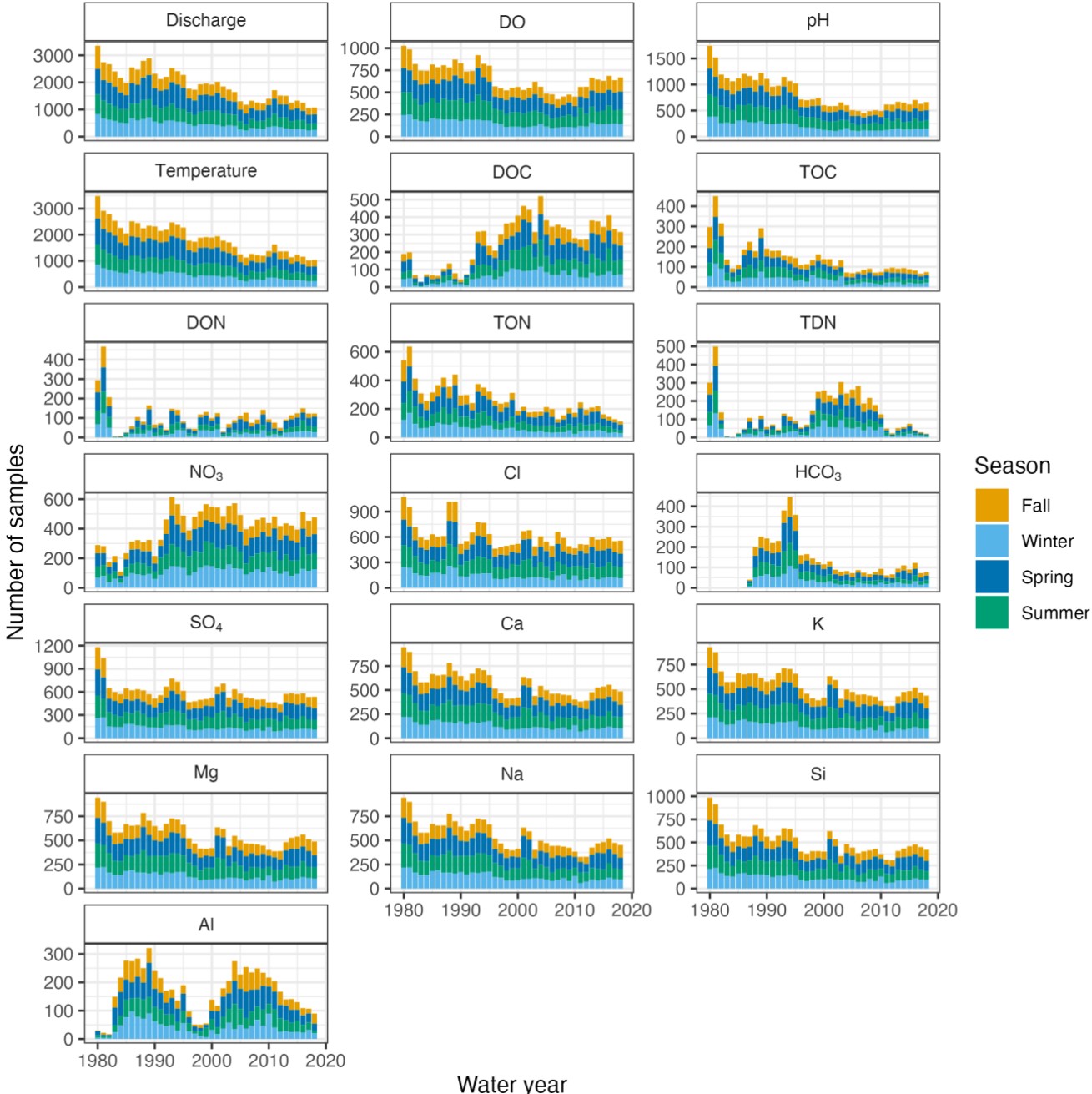

**Figure 4:** The number of samples by water year and seasons (color code) for the stream data between 1980 and 2018 for discharge, dissolved oxygen (DO), pH (field), temperature, dissolved organic carbon (DOC), total organic carbon (TOC), dissolved organic nitrogen (DON), total organic nitrogen (TON), (i) total dissolved nitrogen (TDN), (j) nitrate ($NO_3$), (k) Cl, (l) bicarbonate ($HCO_3$), (m) sulfate ($SO_4$), (n) Ca, K, Mg, Na, Si, and Al. Note: we do not show alkalinity or DIC here because coverage is so limited in the dataset (Table 2).

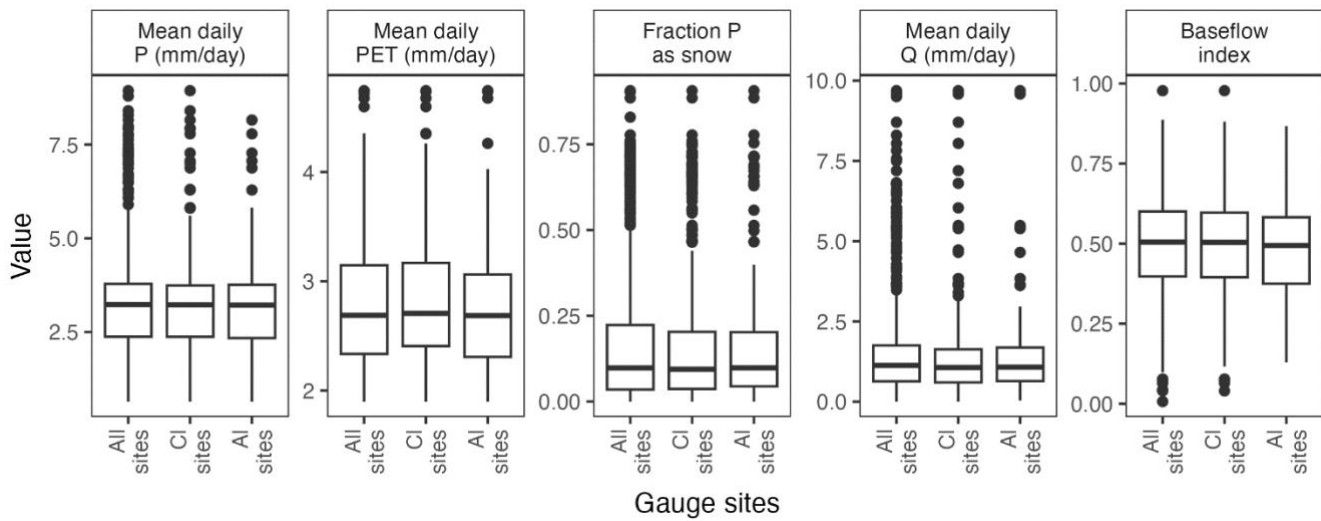


**Figure 5.** Distribution of values for climate and hydrological metrics for all CAMELS gauges ($n = 671$) versus gauges that have chloride (Cl) data ($n = 404$) or aluminum (Al) data ($n = 263$) when the data are subset for water years 1980 to 2018. P = precipitation. PET = potential evapotranspiration. Q = discharge. Boxplots represent the median and interquartile range with outlier values shown as points. See Addor et
al. 2017 for more information on attribute description.

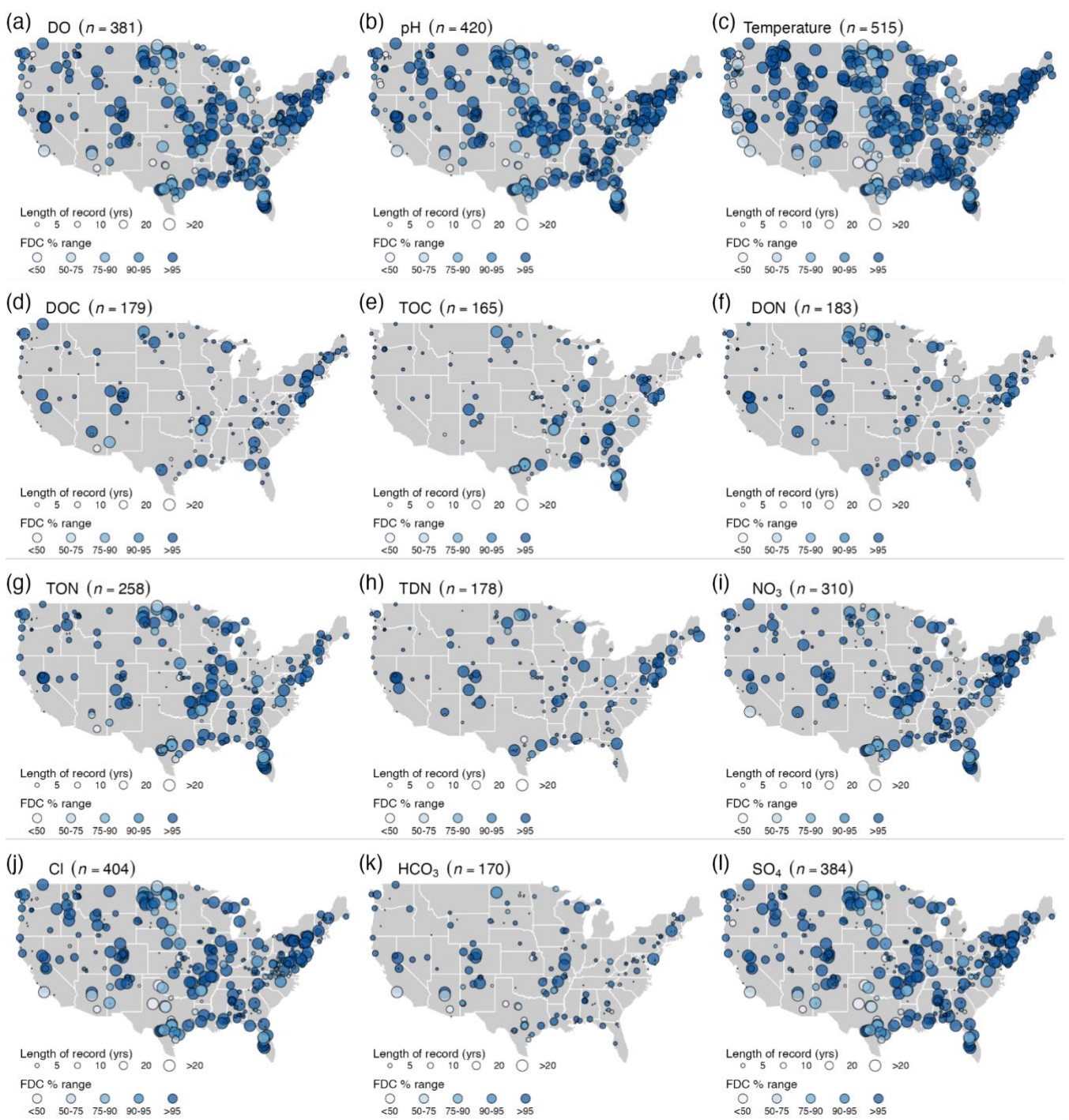

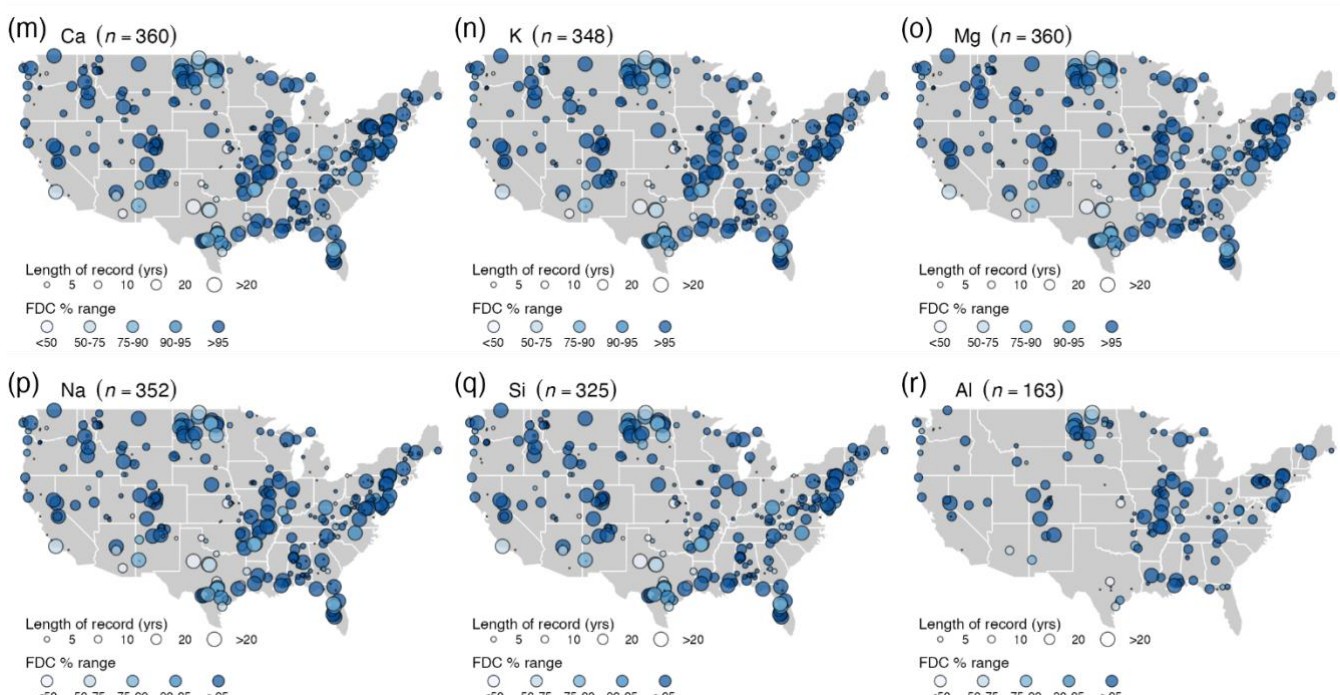

**Figure 6:** Range of the flow duration curve (FDC %, symbol color) and length or record (symbol size) for the stream data between 1980 and 2018 for (a) discharge, (b) dissolved oxygen (DO), (c) pH (field), (d) temperature, (e) dissolved organic carbon (DOC), (f) total organic carbon (TOC), (g) dissolved organic nitrogen (DON), (h) total organic nitrogen (TON), (i) total dissolved nitrogen (TDN), (j) nitrate ($NO_3$), (k) Cl, (l) bicarbonate ($HCO_3$), (m) sulfate ($SO_4$), (n) Ca, (o) K, (p) Mg, (q) Na, (r) Si, and (s) Al. The inset histogram shows the number of samples by 8-year periods. Note: we do not show alkalinity or DIC here because coverage is so limited in the dataset (Table 2).




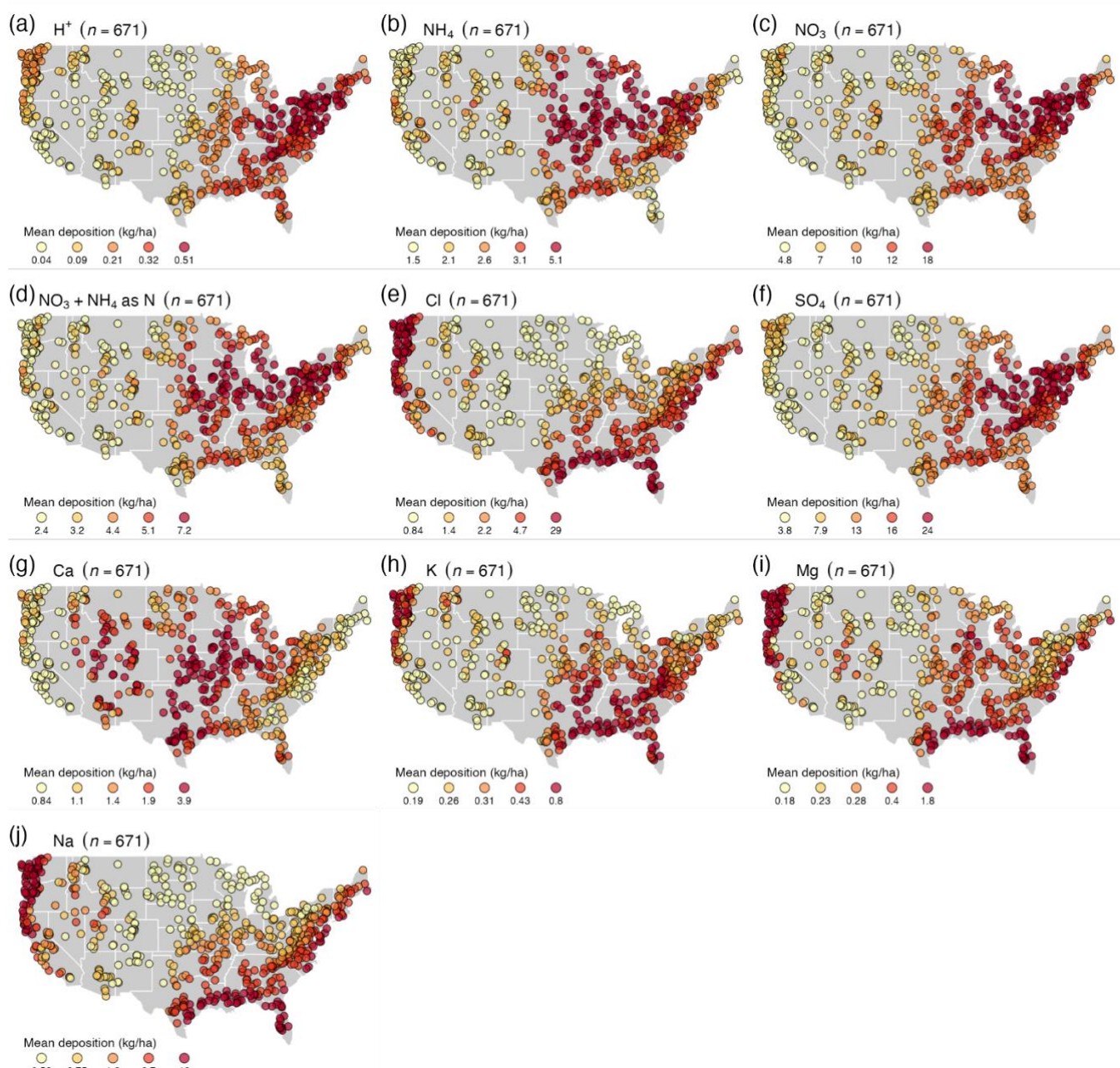

**Figure 7:** Average atmospheric wet deposition in kg/ha (color code) using data from the NADP network from 1985-2018 for (a) protons (H+), (b) ammonium (NH4), (c) nitrate (NH4), (d) inorganic nitrogen (NH4 and NH4), (e) Cl, (f) sulfate (SO4), (g) Ca, (h) K, (i) Mg, and (j) Na. The number of locations represented are referenced as n.

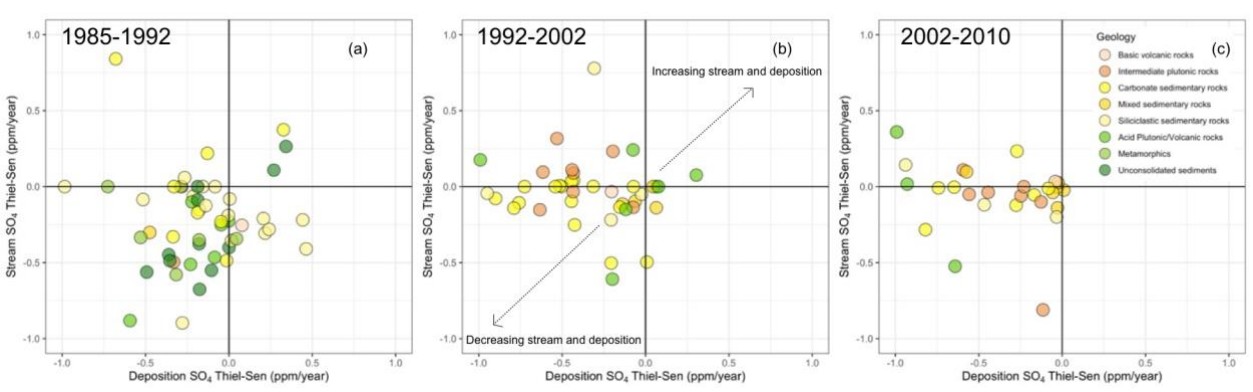

**Figure 8:** Long-term Sen slope (e.g. average trend) from Mann Kendall analysis of $SO_4$ deposition (a) from 1985-1992, (b) 1992-2002, and (c) 2002-2010. Symbol color shows dominant geology.




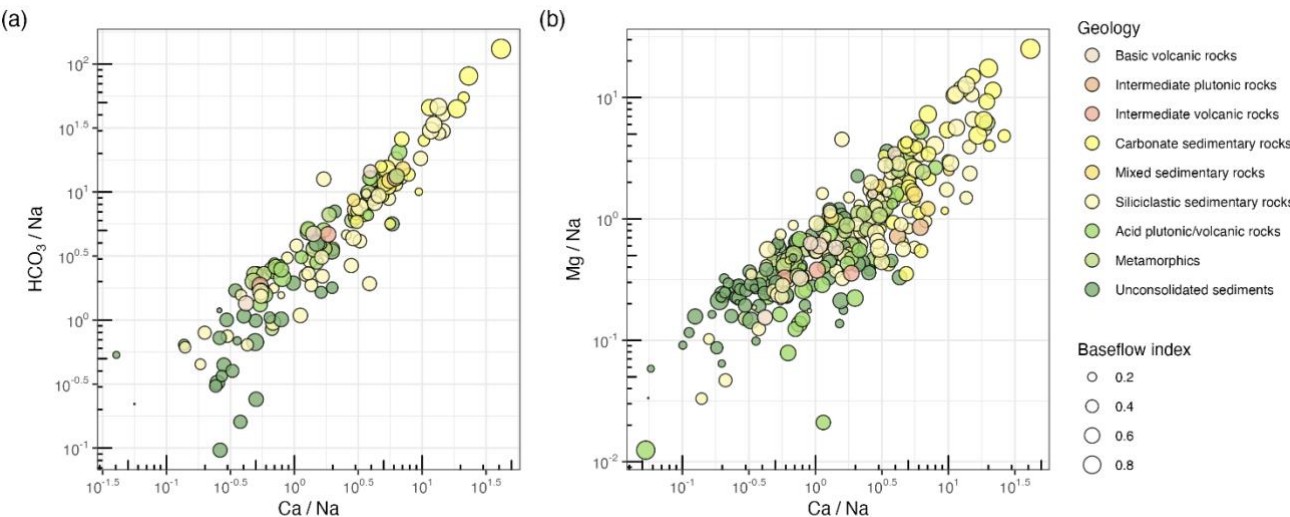

**Figure 9:** Molar ratios of HCO$_3$ and Mg to Na as a function of the molar ratio of Mg/Na at low flows (flow duration curve >66%). Symbol color is the dominant geology and symbol size is the baseflow index.

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
