# Peer review of "CAMELS-Chem: Augmenting CAMELS (Catchment Attributes and Meteorology for Large-sample Studies) with Atmospheric and Stream Water Chemistry Data"

_Hydrology and Earth System Sciences, 2022_

## Author Response (AR1)

**Reviewer 1**

Sterle et al. present a compiled novel dataset of water quality solutes and atmospheric deposition inputs for the CAMELS catchments. Their work augmenting existing and widely used CAMELS datasets is needed for further research in analyzing spatial and temporal water quality trends in minimally impacted watersheds. Existing papers have done large-scale water quality analyses, but few have provided open-access datasets and the breadth of solutes.

At its core, this is a data paper. As such, I think the methods need to be expanded. My comments primarily pertain to the data and methods, as I see that this is the paper's novelty.

The CAMELS dataset is widely used, and the addition of water chemistry provides the opportunity for analysis. However, I think the dataset could be improved substantially. This paper by Sterle et al. provides an excellent resource for the community.

**Authors response:** We thank the reviewer for this assessment and agree that this database presents a great resource for the community and has already found application in research resulting in publications. We are looking forward to seeing how the revised version of this manuscript will be received.

**Reviewer comment:** In the paper, the dataset is stated to end in 2014 (see Data Comments below because I'm not sure if this is accurate). However, in many cases, solute and discharge data are available until the at least end of the NADP reporting period. Soon 2014 will be 10-years ago, and I worry the data will not be quickly obsolete and not used to its fullest capacity. The value of this dataset would be exponential if the authors were able to harmonize data from various agencies.

**Authors response:** We appreciate this comment and agree. We have included data until (and including) the year 2018. We have updated the description where the data range is mentioned.

For example in the introduction, where we state: "Stream data goes back to the early 1900 in some cases and we integrated these data into the dataset. However, since CAMELS covers the timeframe from 1980 to 2014, we showcase stream data starting 1980 and have included the earliest data on atmospheric deposition data by the National Atmospheric Deposition Program (NADP), available since 1985."

**Reviewer comment:** The USGS has developed methods to generate the longest time series possible for watersheds. However, I do not see these methodologies applied to these CAMELS watersheds.

1. The following USGS report looked at the statistical differences between similar solutes. They were then able to merge solutes that were registered under different parameter codes but were statistically similar, allowing for a richer dataset.

2. Secondly, the USGS report also has a methodology for pairing gauges, and sampling locations are not co-located.

3. With the dataRetrieval package in R you can pull USGS, state, tribal, and NGO water quality measurements. This can potentially expand the dataset. However a lot of manual data cleaning is required because of the poor metadata for non-USGS agencies (ex. nitrate can be reported as N or $NO_3$, and not explicitly stated in the metadata) (Sprague et al. 2017). With the data harmonization from multiple sources, I would expect a section on the methodology of data cleaning.

   - Oelsner, G. P., Sprague, L. A., Murphy, J. C., Zuellig, R. E., Johnson, H. M., Ryberg, K. R., Falcone, J. A., Stets, E. G., Vecchia, A. V., Riskin, M. L., Cicco, L. A. D., Mills, T. J., & Farmer,

W. H. (2017). Water-Quality trends in the nation's rivers and streams , 1972 – 2012 — Data Preparation, Statistical Methods, and Trend Results Scientific Investigations Report 2017 – 5006. U.S. Geological Survey Scientific Investigations Report 2017–5006. https://doi.org/10.3133/sir20175006

**Authors response:** We agree with the reviewer that excellent resources exist for expanding data sets, including methodologies that significantly improve common pitfalls such as those mentioned above. We also agree that we did not offer this context for the reader in this manuscript and made changes accordingly. We also were remiss in not citing this important paper and have rectified this issue.

Our objective for this relational database is to supplement water quality and atmospheric data in conjunction with the catchment attributes of the original CAMELS dataset; therefore, for this publication, we did refrain from adding sites and additional harmonization. However, this will offer opportunities for further expansion and analysis of this database going forward.

However, we agree on merging solutes when possible and to clarify what parameter codes are included (Table 1). For example for chloride, sulfate and a nitrate, data from unfiltered samples can be used as these ions do not have a particle form and are dissolved independent of filtering. We have included this in the table.

**Reviewer comment:** Figures capture the various hydrological/biogeochemical metrics for select solutes in section 4. However, as a user, I find it challenging to evaluate whether the data would be sufficient for my use case. The authors have included some analysis of data coverage in section 3.1, however, since the strength of this paper lies in the data there could be more information to help users evaluate whether this dataset suits their needs.

**Authors response:** We agree with the reviewer that we chose a somewhat unconventional way of presenting data and possible applications and that both can be improved. We therefore have added new sections (3.2, 3.3, and 3.4) and comprehensive maps for all solutes for the reader to be able to evaluate the dataset. This includes information on the consistency of sampling across time, seasons, and discharge and the co-occurrence of samples. Additionally, we show maps on data availability over the time frame that CAMELS and stream solute data are available (since 1985). These include the number of samples and length of records, mean concentrations and coefficient of variation, atmospheric deposition for all variables, % range of flow duration curves, and maps of the 66 percentile of flow duration curve. Further, in supplementary materials we provide all data (since beginning of the record).

The example applications are intentionally included (with the removal of the previous example 2) to inspire use of this dataset by a broad range of disciplines that otherwise might not consider using these data. Especially interdisciplinary fields, such as Ecology and Critical Zone science that are in need of data that capture the system from "tops of the tree canopy to the bedrock and groundwater", to investigate how atmospheric drivers interact with the critical zone to impact dissipative products e.g. as stream waters. Our examples showcase an array of example applications to facilitate these types of data-base approaches. As such, our manuscript is a hybrid of a paper that accompanies the dataset, but also an outreach tool for communities to use. We added this framing to L. 24-25" To motivate use by the larger scientific community across a variety of disciplines, we exemplify the potential use of this dataset for three analyses", and L. 200-201" To motivate adoption of this dataset by the larger scientific community, especially in fields that span multiple disciplines, we intentionally decided to offer examples in three themes:".

**Reviewer comment:** There should be summary figures for all solutes, so users can adequately assess whether the dataset is appropriate for their use. I think the paper would benefit tremendously if the dataset had more metadata and signatures/summary statistics. Specifically, in Addor et al. (2017), the authors

summarized many indices and described the indices in great detail (see Table 2 and Table 3 in Addor et al). I suggest summarizing information about (1) missing data/data gaps, years of continuous data, (2) low/high flow distribution, (3) FDC the WQ spans (figure 7), (4) seasonality of hydrology and solutes, and other metrics that the authors deem useful.

**Authors response:** We appreciate the reviewers comments that are geared towards making this database usable. In addition to the comprehensive maps described above, we now provide requested summary statistics in new tables (Table 3) that we integrated into the main manuscript and supplementary materials.

*These comments pertain to the files on the google drive.*

**Reviewer comment:** Data provided has inconsistencies in the way the date is reported. Example from camel_chem_v3.

- Gauge ID 14309500: sample_timestamp reads 8/15/67 15:00 whereas other dates are in 2010-12-11 format.
- Gauge ID 6447000: sample_start_dt appear to indicate that the data from 1950s. If so, this falls outside of the time period listed in the paper.

**Authors response:** We thank the reviewer for this comment and have made the changes as requested.

**Reviewer comment:** It appears that the data available in camel_chem_q_v1 ends in 2018. Please update the manuscript with the correct dates if this data is available.

**Authors response:** We clarified in the MS that the data of the database is updated regularly to include more recent data. However, for the Figures and summary statistics offered in this publication we use data until and including the year 2018.

**Reviewer comment:** The original CAMELS dataset provides Shapefiles, however, to allow for seamless merging of data, the header used to identify which column the watershed IDs are the same as the name used by the original CAMELS dataset.

**Authors response:** we have ensured seamless merging of our data with the original CAMELS dataset. We have used the same column name for the watershed IDs in our dataset as the name used by the original CAMELS dataset, which is "gauge_id". This should allow for easy integration and analysis of our data with the original dataset.

**Reviewer comment:** The dataset provided should be able to stand on its own without needing the other CAMELs dataset. The watershed metadata should thus be included (area, outlet latitude and longitude, USGS gauge number with leading 0s).

**Authors response:** Thank you for your feedback on our dataset. We agree that this is an important consideration and have made efforts to ensure that our dataset contains all necessary metadata.

In response to your specific comment about the inclusion of watershed metadata such as area, outlet latitude and longitude, we would like to confirm that these variables are indeed included in our dataset. The columns

"area_gages2", "gauge_lat", and "gauge_lon" contain the area, outlet latitude and longitude information respectively.

**Individual scientific questions/issues ("specific comments"),**

**Reviewer comment:** Line 84-87: Authors state that they have WQ data from 1980-2014. Later the authors state that the data is "for the same time period" as NADP data (1985-2019). Dates should be consistent.

**Authors response:** we agree and have revised this throughout the MS. The data we showcase ranges from 1980 to 2018. However, the NADP data is only available since 1985, thus we clarified "(...) since CAMELS covers the timeframe from 1980 to 2014, we showcase stream data starting 1980 and included the earliest atmospheric deposition data from a product developed by the National Atmospheric Deposition Program (NADP, available since 1985)".

**Reviewer comment:** Line 103: Section 2.2 should be written in a high-level abstract way. I find it unclear how these frameworks are applied to this data in the way it is currently written. I would be better to see more specificity and names of the database (ex. NADP, NWIS).

**Authors response:** Thank you for this suggestion. We have rewritten the section on our Data Harvesting and Integration approach (Section 2.2) to more directly speak to why and how the Extraction, Transformation and Loading framework was applied to generate the CAMELS-Chem relational database. We also moved the order of this section to follow a presentation of the sourcing and characteristics of the specific data sets integrated within CAMELs-Chem (i.e., NWIS, NADP)

**Reviewer comment:** Line 125: It is unclear whether "daily average discharge" means a continuous dataset or just discharge measurements for the data that there are solutes. The dataset provided suggests the latter, but I think there would be value in providing the daily discharge timeseries for the same timespan of the solute data.

**Authors response:** The daily discharge we reference are the USGS average discharge values. The instantaneous data reference hourly or finer time scales. The continuous daily discharge data are part of the original CAMELS datasets, and are therefore easily available.

**Reviewer comment:** Line 135: In Table 2, deposition units are reported as mg/L. However, NADP reports their deposition in both concentration and kg per hectare. Are the units in Table 2 a mistake? If not, can you add some detail on the methods used to convert concentration to an area normalized load?

**Authors response:** we agree with the reviewer and have reported both the concentration data and the deposition units as kg/ka in Table 4 .

**Reviewer comment:** Table 1: Consider added the NWIS parameter code. For example, is "Nitrate, water filtered" the nitrate plus nitrite (00631) or just nitrate (00618)? There are many parameters for slightly similar solutes and it would help with reproducibility if the parameters codes were included. Also, consider listing the difference between pH in the field and pH in the lab for users.

**Authors response:** We agree with this suggestion and have included the parameter condes to the table. Further, we were able to increase the number of observations for Cl, nitrate and sulfate by adding unfiltered samples as these are species where filtration does not impact concentration. To help the user we added all of this information, including column abbreviations, to the table.

**Reviewer comment:** Table 1: Also consider adding more detail to units. For example, is nitrate mg-NO3/L or mg-N/L.

**Authors response:** We agree and added this information for nitrate (mg/L N) to the Table. For all other solutes the units are not ambiguous but we provide the parameter codes as requested.

**Reviewer comment:** Line 204: EPA link is broken. I have had many issues with direct links where they are archived and become a dead end. I highly encourage the authors to find a paper with a DOI to support this sentence. As a starting point, you can consider:

- Baumgardner, R. E., Lavery, T. F., Rogers, C. M., & Isil, S. S. (2002). Estimates of the Atmospheric Deposition of Sulfur and Nitrogen Species: Clean Air Status and Trends Network, 1990−2000. In Environmental Science & Technology (Vol. 36, Issue 12, pp. 2614–2629). https://doi.org/10.1021/es011146g
- Lloret, J., & Valiela, I. (2016). Unprecedented decrease in deposition of nitrogen oxides over North America: the relative effects of emission controls and prevailing air-mass trajectories. Biogeochemistry, 129(1-2), 165–180.

**Authors response:** We agree with the reviewer and have replaced the weblink with references as suggested.

**Technical corrections ("technical corrections": typing errors, etc.).**

**Reviewer comment:** Subscripts for solutes should be consistent throughout the manuscript.

**Authors response:** We agree and have fixed the subscripts for $NH_4$, $NO_3$, $SO_4$ etc throughout the MS and supplementary materials.

**Reviewer comment:** Table 3 formatting caused solutes to be cut off.

**Authors response:** We have reformatted the table.

**Reviewer comment:** Line 18 and 149-151: 18 solutes listed in the abstract, 17 listed in Table 1 and in Line 149, and 16 listed in the text. Make them all consistent.

**Authors response:** We agree that this is confusing and stemmed from the omission of temperature in the original table. We have rectified the issue and revised with the correct # of solutes (18) throughout.

**Reviewer comment:** Line 57: Remove (?).

**Authors response:** Revised as suggested.

**Reviewer comment:** Line 60: Remove "CITE"

**Authors response:** Revised as suggested.

**Reviewer comment:** Line 168, 174 and others: When referencing figure (ex Figure 2), please add the panel letters (a,b,c, etc.).

**Authors response:** We agree and added the panel letters as requested for all figures throughout the ms.

**Reviewer comment:** Figure 2: Panel a, why does daily average discharge only have 393 watersheds while the original CAMELS dataset uses USGS discharge in the original CAMELS dataset?

**Authors response:** we agree that this is confusing and we clarified in the ms. The discrepancy is coming from the fact that not all gauges have paired Q and water quality data available for the time frame we report in this paper. For the timeframe of the dataset (1985-2018) and of the original 671 CAMELS watersheds, 506 watersheds returned instantaneous or daily mean discharge data and 488 returned both discharge and water quality data.

**Reviewer comment:** Figure 6: Regarding NO3, if arid and humid sites are a subset of all sites, I am unsure how the slope for all sites can be larger than both arid and humid.

**Authors response:** We agree with the reviewers and found that this example does not add substantially to the Manuscript. We decided to remove this figure from the MS.

**Reviewer 2:**
This paper describes efforts to amend the known and quite successful US-wide CAMELS dataset with hydrochemical and deposition data. This effort is overdue and would greatly enhance the usage of the original CAMELS data as well.

**Authors response:** We thank the reviewer for this assessment and agree that this database presents a great resource for the community and appreciate the comments that are geared to making this a better and more user-inspired resource. We are looking forward to seeing how the revised version of this manuscript will be received.

**Reviewer comment:** Having stated this I have to admit that the manuscript is not convincing to me. It fails to state what data (e.g., constituent codes in the USGS data) was used in what exact way (missing description of data evaluation, conversion, filtering). It visualizes data coverage but does not state number of observations and number of stations in a consistent way. So overall the reader is left rather unclear about the whole data handling process and the outcome.

**Authors response:** We agree that the user will need this information and have added the parameter codes to the table and also offered more context for data evaluation and processing. We further clarified how many gauges are included as "n" in all figures and offered additional maps for all solutes (Figure 2, 3, 4, 7 and 9).

**Reviewer comment:** While I like the idea of three examples this data could be used for, this leaves me rather puzzled. This is submitted as a research paper but does not really come up with research. For me this manuscript would rather fit the purpose of ESSD (Environmental Science System Data) as a dataset description than HESS as a research paper. For the latter it would have an appropriate structure and depth but still would need to acknowledge the detailed comments below.

**Authors response:** We agree with the reviewer that we chose a somewhat unconventional way of presenting data and possible applications that bring us to a gray area between a data paper and a research paper. Working in an interdisciplinary group with domain and data scientists has repeatedly confronted us with the disconnect between domain knowledge across multiple disciplines and database science. We therefore made the intentional decision to go beyond a traditional data paper by offering examples that demystifies the application of such a relational database especially for interdisciplinary fields such as ecology and critical zone community. These examples help illuminate how these data can be used and therefore facilitate the integration of ideas and data from different disciplines. This is essential for generating new ideas as it is often challenging to see connections across disciplinary boundaries.

Especially interdisciplinary fields, such as Ecology and Critical Zone science are in need of data that capture the system from "tops of the tree canopy to the bedrock and ground water", to investigate how atmospheric drivers interact with the critical zone to impact dissipative products e.g. as stream waters. Our examples showcase an array of example applications to facilitate these types of data-based approaches.

However, we agree that we failed to offer enough information that a typical data publication has to offer and revised our presentation accordingly for user evaluation. We have added new sections of text (3.2, 3.3, and 3.4) comprehensive maps for all solutes for the reader to be able to evaluate the dataset. This includes maps on data availability (number of samples and length of records, Fg. 2), mean concentrations and coefficient of variation (Fig. 3), atmospheric deposition for all variables (Fig 4), % range of flow duration curves (Fig. 7), and maps of the 66 percentile of flow duration curves (Fig 9), as well as additional summary statistics.

**Reviewer comment:** The abstract needs to transport the content and motivation in a better way. It misses quite some information e.g. resolution of provided data. It is not clear what is really provided in this new data set and what is already available from CAMELS.

Line 18: The linkage to the original US CAMELS dataset remains unclear in the abstract. Is this an addition to the original one as induced by title and text here or is this something completely new.

**Authors response:** We agree with the reviewer that we did not provide enough background and framing for the reader in the abstract and appreciate this comment. Indeed, CAMELS-Chem is both an amendment of existing CAMELS dataset and also something new, as the relational nature of this dataset allows a completely new set of applications. The relevant section in the abstract now reads:"Large sample datasets are transforming hypothesis testing and model fidelity in the catchment sciences, but few large stream water chemistry datasets exist with complementary streamflow, meteorology, and catchment physiographic attributes. The existing CAMELS (Catchment Attributes and Meteorology for Large-sample Studies) dataset includes data on topography, climate, streamflow, land cover, soil, and geology across the continental U.S. With CAMELS-chem, we now pair these existing catchment attribute data with atmospheric deposition data from the National Atmospheric Deposition Program and water chemistry data and instantaneous discharge from U.S. Geological Survey over the period from 1980 through 2018 (...)"

**Reviewer comment:** Line 24: This is odd. Either give the exact number or >. Rounding to something easier to read would probably best.

**Authors response:** We agree and have removed ">" which was an editing error.

**Reviewer comment:** From my point of view the introduction should state some examples of existing water quality databases. There are recent advances here such as: GRQA: Global River Water Quality Archive (Virro et al., ESSD), GLORICH (Hartmann, J., Lauerwald, R., and Moosdorf, N.: A Brief Overview of the GLObal RIver Chemistry Database, GLORICH, Proced. Earth Plan. Sc., 10, 23–27, https://doi.org/10.1016/J.PROEPS.2014.08.005, 2014. a, b) or QUADICA (https://doi.org/10.5194/essd-2022-6)

**Authors response:** We were indeed remiss in referencing these datasets and have rectified this oversight in the current version. We have added reference of these datasets to L. 43 ff. Thai section now reads "The Global River Water Quality archive (GRQA) and GLObal RIver Chemistry Database (GLORICH) have offered opportunities for water quality analyses across time and land scale (Hartmann et al. 2014, VIrro et al. 2021) and the Catchment Attributes for Large-Sample Studies (CAMELS, Newman et al., 2014) compile high quality streamflow measurement in 671 unimpaired catchments, as well as climate forcing datasets

(e.g. daily precipitation and temperature) and physiographic properties (e.g. land cover, topography, etc., Addor et al., 2017). CAMELS has seen widespread adoption by the hydrological community as a benchmarking tool for hydrological models (Melsen et al., 2018; Mizukami et al., 2019; Pool et al., 2019), in the development of hydrological signatures and new information theory-based approaches, and the application of novel machine learning tools (Kratzert et al., 2018)." The combination of catchment attributes and generation of matching datasets on stream water chemistry has recently been developed for Germany (Ebeling et al. 2022), however, for the CONUS this approach has not  not seen as much development. Furthermore, atmospheric deposition data is typically not included in such data sets, despite the significant impact of atmospheric contribution to stream chemistry."

**Reviewer comment:** Line 31: There are quite a number of examples of nation-wide to continental scale water quality studies using more than a single catchment. I disagree here that availability of datasets does not go hand in hand with usage of this.

**Authors response:** We actually meant to emphasize the challenge of analyzing processes across bio-geo-hydrosphere where single sites often have data, but might not have the breadth for regional, continental or global studies. Essentially it is the problem of regional and continental patterns vs. site specific process identification and "uniqueness of place", however, we realized we did not frame this well. We revised accordingly, the first sentence now reads: Earth surface processes include coupled and complex processes that involve atmosphere, biosphere, lithosphere and hydrosphere, however, tracking these important processes across time, space and disciplines remains a challenge that is, amongst others, related to data availability and data connectedness."

**Reviewer comment:** Line 35: Check citation formatting here.

**Authors response:** revised as requested

**Reviewer comment:** Line 35: Can you be more specific on the "issues" mentioned here?

**Authors response:** We clarified that the uniqueness of place issue is the disconnect between catchment specific processes vs larger scale patterns. The section now reads: "One of the key advantages of aggregating and harmonizing data into larger sample size datasets is to test how model hypotheses reproduce observed behavior across variable conditions and sites to reduce the uniqueness of place-based issues that plague catchment science (Gupta et al., 2014, Hubbard et al., 2020), that is, individual (unique) catchments might not be generalizable to explain a larger scale pattern and vice versa".

**Reviewer comment:** Line 42: Year in citation missing.

**Authors response:** Fixed as requested.

**Reviewer comment:** Line 43: Use the complete name of CAMELS here.

**Authors response:** We agree and introduced the abbreviation at first use here.

**Reviewer comment:** Line 56: Check the question mark here.

**Authors response:** This was an editing error that we have removed.

**Reviewer comment:** Line 60: Check citation here.

**Authors response:** Fixed as requested.

**Reviewer comment:** Line 83: I think you should better introduce the idea to provide atmospheric deposition data.

**Authors response:** We agree and have provided framing in the introduction: L. 30ff "Earth surface processes include coupled and complex processes that involve atmosphere, biosphere, lithosphere and hydrosphere, however, tracking these important processes across time, space and disciplines remains a challenge that is, amongst others, related to data availability and connectedness" and L. 50 ff:" The combination of catchment attributes of matching datasets on stream water chemistry has recently been developed for Germany (Ebeling et al. 2022), however, for the CONUS this approach has not not seen as much development. Furthermore, atmospheric deposition data is typically not included in such data sets, despite the significant impact of atmospheric contribution to stream chemistry".

**Reviewer comment:** Line 86: Can you specify why to stop in 2014? Because the original CAMELS data also have that time frame?

**Authors response:** We agree that this requires clarification. The CAMELS catchment attributes timeframe spans 1980 to 2014, but our stream data goes back to the early 1900 in some cases. The NADP data starts from 1985. We include all data in the dataset and feature it in supplementary materials, however in the main body we showcase data for the time where CAMELS and stream data is available (since 1980). We clarified thai in the introduction and also methods. Further, the database can be readily updated with more recent data, but we chose a cutoff for the data we showcase in the publication. We now show all data until and including the year 2018.

**Reviewer comment:** Line 94: I would expect a citation to the NWIS data sources mentioned here.

**Authors response:** We agree and have added the citation

**Reviewer comment:** Line 97: What is meant with the "two datasets"? "For each observation" seems unnecessary here.

**Authors response:** We have clarified that we mean two data sources and have removed "for each observation". The section now reads "NWIS data also have unique gauge identifiers for each observation reducing the complexity of merging the two data sources".

**Reviewer comment:** Line 97: The match with the CAMELS station does not explain the promised "geographical coverage". Please adjust this sentence. An option could be a reference to a figure showing the distribution of stations, maybe including the ones from the CAMELS data that are not amended with chemical data.

**Authors response:** We agree and have included maps to showcase the capacity of CAMELS-Chema across a number of attributes (e.g. Figure 1, 2, 3, 4, 5 and 6).

**Reviewer comment:** Line 102: From my point of view the 2.2 chapter on the methods is not informative and needs some revision. The reader does learn some fancy names but not what exactly was done with the data. The linkage of methods to the provided text-files remains unclear. This does not fit the previously stated wish to make everything reproducible by providing scripts. See also details below.

**Authors response:** Based on this comment and similar ones from Reviewer #1, we have rewritten a section on our Data Harvesting and Integration approach (Section 2.2) to more directly speak to why and how the Extraction, Transformation and Loading framework was applied to generate the CAMELS-Chem relational database. We also moved the order of this section to follow a presentation of the sourcing and characteristics of the specific data sets integrated within CAMELs-Chem (e.g., NWIS, NADP).

**Reviewer comment:** Line 103: I cannot follow this argumentation.

**Authors response:** These lines have now been replaced by the rewritten section.

**Reviewer comment:** Line 109: Is ETL an established method? If so, please reference this. If not: Is there really a need for a fancy name for a standard data transformation method needed that uses a scripting language in a reproducible way?

**Authors response:** We added more information: in section 2.2.: Extraction Transformation and Loading is a framework that has been established in the business information world for some decades (Kimbal R, Caserta J. 2004. The Data Warehouse ETL Toolkit. USA: Willey Publishing, Inc), but only more recently has it been applied in environmental data management. In our rewritten section, we focus less on ETL as a novel method, and instead provide specifics of why and how the ETL framework was applied to generate the CAMELS-Chem database.

**Reviewer comment:** Line 112: I do not see the link from this method to the resulting data provided, which are easy-to-use but simple text-files. This paragraph seems to be overly complicated.

**Authors response:** These lines have now been replaced by the rewritten section.

**Reviewer comment:** Line 116: Similar as before. Either reference this as established methods or leave it. These abbreviations are not used further in the manuscript - so why introducing them here?

**Authors response:** We agree. These lines have now been replaced by the rewritten section.

**Reviewer comment:** L131: "once per day" at the same time? Is this what is meant here?

**Authors response:** We agree that this sentence needed clarification and have reworded "We address this by making the assumption that field technicians generally collect samples for multiple solutes at the same time."

**Reviewer comment:** Line 135: Again a reference to the dataset would makes sense for me. It would be, moreover, helpful for the readers to know about the absolute number of stations the interpolation is based on.

**Authors response:** We agree and have added this information to the MS (e.g. in abstract and methods).

**Reviewer comment:** L 137: I do not understand what "align" means in this context. Please be more specific.

**Authors response:** We reworded this section for clarification: Data rasters were positioned to correspond to CAMELS catchment shape files to determine total watershed deposition for 10 species for a given year.

**Reviewer comment:** Line 139: Somehow strange to mention Table 2 first in the text while Table 2 comes second.

**Authors response:** We agree that this formatting is not helpful and made sure that table 1 is referenced before table 2. For the final version the Journal will clarify the formatting and placement of tables.

**Reviewer comment:** Table 1: It would be helpful, if not done elsewhere, to state the USGS parameter codes as this often causes confusion. The caption is not fully matching the column names, e.g. is abbreviation=attribute? Also it would be good if this is also consistent with or clearly linked to the column names in the provided data files, which seems to be not the case.

**Authors response:** We agree and have added all USGS parameter codes and abbreviations to the table.

**Reviewer comment:** Line 145: The "+" seems unnecessary.

**Authors response:** Fixed as requested

**Reviewer comment:** Line 151: This description says all what is needed. No further need for a table here.

**Authors response:** We can see what the reviewer means, but prefer to keep the Table so that the reader has a quick comparison at hand on what parameters are included for atmospheric inputs vs water quality.

**Reviewer comment:** Line 151: Please use subscript in the constituent's abbreviations similar to the table where the constituents are introduced. This applies to the entire manuscript.

**Authors response:** We agree and have revised throughout.

**Reviewer comment:** Table 3: I do not get the meaning and idea of Table 3.

**Authors response:** We agree, it was redundant and we have removed the table.

**Reviewer comment:** Line 164: This chapter would profit from a table (maybe combined with table 1) that lists number of stations and number of observations and maybe median number of observations per station for each of the constituents.

**Authors response:** We agree that the number of observations is a very important component for user evaluation of the data. We chose to supply this information with our maps, which now include all solutes.

**Reviewer comment:** Line 171: I was not aware of this varying foci. Is there a reference for that?

**Authors response:** The shift in foci is localized in some cases and we have provided a reference "CAMELS-Chem offers long-term records for trend analysis and broad geographic coverage in catchments (Figure 3 as well as S4 in supplementary materials). Because USGS sampling foci varied between decades, temporal biases in the sampling record exist (Shanley et al. 2015). ".

**Reviewer comment:** Line 175: This sentence needs to be checked - the logic is not clear.

**Authors response:** We agree and clarified:"The $NO_3$ data are more abundant in the Midwest and east coast where agriculture is generally more intense and focussed sampling for nutrients is common.

**Reviewer comment:** Line 186: While the previous chapter states reasons for different data density and distribution. Here, however, suddenly spatial pattern in the deposition is described. This belongs elsewhere. The chapter should describe the meta-information only.

**Authors response:** We agree and have revised the entire section. Atmospheric deposition is now under a new header as section 3.5. "Atmospheric deposition needs to be considered when evaluating water chemistry patterns and especially for weathering studies, the contribution of atmospheric deposition needs to be corrected for (Berner and Berner 2012). For example, Cl and Na deposition values are higher in coastal areas (Figure 7e and j), while $NH_4$ and $NO_3$ deposition vales are higher in places where anthropogenic inputs of fertilizer are high (Figure 7b and c). Ca typically has higher values away from coastal areas and is strongly impacted by local bedrock and soil composition (Berner and Berner 2012). In many cases these patterns are consistent with patterns in stream chemistry (e.g. Figure 1j for stream $NO_3$)".

**Reviewer comment:** Line 226: "not" missing?

**Authors response:** Agreed, we fixed this section:"As expected, plotting trends in $SO_4$ stream chemistry and wet deposition for an earlier timeframe (1985-1992, Figure 8a) decreasing trends in $SO_4$ deposition and corresponding decreasing trends in $SO_4$ stream chemistry are apparent. Wet deposition trends remain decreasing in the following two decades (1992-2002 and 2002-2010), but without much response in $SO_4$ stream chemistry".

**Reviewer comment:** Line 234: This reads quite strange. You compute cq relationships to check if there are cq relationships? Consider revising.

**Authors response:** We agree and this has been fixed through rewriting the entire section on C-Q relationships (now section 3.3).

**Reviewer comment:** Line 250: I do not get the meaning of "overlapping solutes" here.

**Authors response:** We agree and this has been fixed through rewriting the entire section.

**Reviewer comment:** Line 251: Isn't this part of the A (attributes) in the CAMELS dataset?

**Authors response:** Yes, we agree that we have not set this up well earlier in the description. The way we designed CAMELS-Chem is that all catchment attributes are now connected to these additional datastreams and can be queried as such. This is actually the main contribution of the database.

**Reviewer comment:** Line 127: I very much like the analysis of FDC coverage. For me this is, however, not just an addition to the cq analysis but rather part of the dataset description in the former chapters.

**Authors response:** We agree and have added the FDC analyses as part of the dataset description and include maps in Fig 3 and Fig. 6.

**Reviewer comment:** Line 269: "response of the relationship"? Consider revising.

**Authors response:** We agree and have revised "the slope" of the concentration-discharge relationship.

**Reviewer comment:** Line 270: What are uneven collection dates?

**Authors response:** We agree and this has been fixed through rewriting the entire section

**Reviewer comment:** Line 299: This statement seems to be unrelated to stream chemistry and thus not directly a result of CAMELS-Chem.

**Authors response:** We agree and this has been fixed through rewriting the entire section

**Reviewer comment:** Line 302: This sentence is very long. What exactly is meant by large-scale response? The observed response is always local, isn't it?

**Authors response:** We agree and this has been fixed through rewriting the entire section

**Reviewer comment:** Line 404: This sentence is redundant, please revise.

**Authors response:** We have removed this sentence.

**Reviewer comment:** I think the authors may already give an idea which repository they are aiming at.

**Authors response:** We have decided to make the dataset (1980-2018) available on Hydroshare and a doi is available and included in the MS. The database is housed at the University of Vermont as it will be continuously updated.

---

## Author Response (AR2)

**Response to Reviewers**

Two reviewers (including one from the previous round) have evaluated your revised manuscript and provided detailed comments. While they both agree that significant improvements have been made compared to the original version, they still find the manuscript has some shortcomings. Both reviewers noted a lack of details in the Methods and Results sections (mostly sections 2 and 3), which led them to comment (negatively) in different ways: one found that the lack of details made your work not reproducible, while the other felt that your dataset, as described, may appear as being no more than a "merge of USGS flow and water quality data", which is something that can be done using an existing R package (and hence is not really novel or value-added). Obviously, the CAMELS-Chem dataset is (or has the potential to be) *much more than a simple data-merging exercise*, but the manuscript in its current state is underselling that potential (I do share the reviewers' opinions on that). One of the reviewers sees your paper as mostly a data paper while the other thinks that submitting your paper to HESS (rather than Earth System Science Data (ESSD), for instance) means that the goal of the research should go beyond just the presentation of the dataset. Personally, I think that your paper would be most impactful as a hybrid one between those two formats, but this makes it tougher to write because you need to articulate not only technical details pertaining to the creation of the dataset, but also hydrological and earth system process details. Both reviewers have included numerous comments for your consideration, so I invite you to carefully look at them and see how you could address them. I do look forward to receiving and reading your revised manuscript and response document, which will be sent out for another round of review.

With best wishes,

Genevieve Ali

**Dear Dr. Ali,**

**We appreciate your thoughtful feedback on our revised manuscript and the patience of the reviewers to improve the manuscript. We have worked hard to increase the research relevance of the manuscript by 1. Including research questions, 2. More clearly answering those questions, and 3. Giving more examples in Section 4 to highlight the broad research applications. We believe the newly revised manuscript fits the sweet spot between a research article and a dataset release.**

Reviewer #2

General Comments

The authors describe an extension of the well-known CAMELS data set with water chemistry and deposition data. I was reviewing a first version of the manuscript already and still think that this is a super-valuable data compilation. However, the manuscript still has, from my point of view, some serious shortcomings. My expectation is that the manuscript shows how data was merged and gives an overview on the dataset. The method is, to my surprise, not reproducible and remains very vague. The results are mostly displayed in a visual way - I very much miss a quantitative overview on the basic descriptive statistic, on the sample frequency per year. I also miss a brief overview on what information are delivered with the original CAMELS data set. So, it is hard for the reader to say if the dataset actually fits his or her needs. Finally, I have problems with the embedded two examples. Motivation and methods are given in chapter 4 and are not aligned with the introduction and methods of the entire manuscript. Results of that examples are not much more than two plots and the statement that others may interpret that. So, overall I think this manuscript needs some serious revision.

We appreciate the reviewers thoughtful comment on the manuscript. We have added considerably to Section 2 to improve the reproducibility of the results. Section 4 is also better aligned with the introduction and methods are made more clear.

Specific Comments
Abstract
L20: Potentially remove the first "and" here.
Edited

L21ff: I cannot fully follow which constituent is given with full name and which is not - consider to homogenize that. Nitrate is given first as a full name, then as the short chemical form and then again as full name.
This was made more consistent.

Introduction
L45: GLORICH (among other water quality databases) is part of GRQA. Not sure how to address this here but I encourage you to make this more clear in this sentence.
Changed

L50-55: You mix water quantity data efforts such as CAMELS with water quality AND water quantity data efforts such as Ebeling et al. here. I think this need better separation and clarification. Why should an atmospheric deposition dataset be relevant for water quantity? Note that atmospheric deposition of nitrogen is part of Ebeling et al.

Added to line 60: "Furthermore, atmospheric deposition data is available for CONUS but has seen less inclusion in such data sets, despite the significant impact of atmospheric contribution to stream chemistry (Shao et al., 2020). "

L75: There is something wrong here. Please check the whole sentence.
Corrected

L85: Be precise - do you mean stream "flow" data?
Yes, corrected.

L87: This is a bit unclear. Why 2018, when CAMELS ends in 2014?
We are trying to maximize the available data.

L90ff: As already stated in the first review round I am not convinced to publish this as a research paper if the manuscript does not attempt to set itself a scientific goal. I think the two examples needs a motivation within the introduction section already and should be mentioned explicitly. The examples are not totally randomly selected as written later in chapter 4!
We understand this criticism and have added research questions that are more clearly dissected in Section 4.

Materials and Methods
Please note this section mas majorly edited.

L105: That is a nice statement but I wonder why it is made here? What is the consequence?
Changed this section completely.

L116: "Water resources data" is not entirely clear for me.
Changed

L96-114: This chapter leaves me a bit puzzled. This is not about "data sources and description". Would there be a more fitting header? Where are actual multiple data sources described?
Majorly edited

L126f: I thought discharge data is the key of CAMELS (see also line 47)? Why was discharge not available in all 671 catchments anymore while it was available when CAMELS was published?
There are periods of the CAMELS records without discharge records.

L128: How many catchments had data before and after 1980-2018? Or is this coverage part of the result section?
We focus our discussion on 1980-2018 but add Figure S3 to also give information on the dataset before 1980.

L121ff: The methods should describe you workflow in a way that it is reproducible for the readers. This is not the case. Also matching fig. S1 is not of big help here. I think you need to state what exactly you did to the data. Fig. S1 lists aggregation, filtering, cleansing, profiling, joining and sorting. At least the first three processes are not described in the text.
Agreed.  Section 2 was majorly rewritten.

Results
L141: I would not call discharge a general water quality parameter.
Changed

L143: Also give full names of Cl and the cations.
Not sure we follow this suggestion.  We have consistently use abbreviations such as Cl.

Table 1: You state for some constituents "Water, filtered and total" - is that visible in the data base? Why is magnesium measured in the suspended sediment? This makes it hard to be analysed similar to all other cations. Why is not unit and database abbreviation given for discharge? Why is alkalinity not stated here?
Thanks for catching a mistake with Mg.  Alkalinity was added to this table.

L146: Mention the fig. S5 here specifically. It takes a while finding the right figure when just referring to "supplementary materials"
Added

L148f: Something went wrong here. 325,477 catchments?
Corrected

L155f: Interesting that you start with the spatial distribution. For sure very important. However, does it make sense to start with descriptive statistics on the data? A table such as table 1 but station median, percentiles, number of observations…? Especially as all catchments aim at pristine areas and do not incorporate too much direct human impacts.
We have added what is now Table 3.

Fig. 1: Given the different catchment size, I recommend to give discharge per area (e.g., mm/a) in 1a. Otherwise this is not very informative. I struggle with the colors vs. size. More intuitive for me would be the value represented by colors and the CV by symbol size. However, this is up to you. For pH it does not work well - but here you can maybe scale the symbol size between min and max similar to the other plots.

We agree that colors versus size is in the eye of the beholder. We elected not to change the discharge units.

Fig. 2: Why is this coming after fig. 1? Seem to be a better fit for the methods or an earlier point in the results.

We switched figure 1 and 2.

L158: These are maybe not the best examples as DO is having strong physical constraints and pH is a log-unit.

Agreed, these were changed to Cl and Na.

L183ff: From point of view of the USGS these are impressing numbers. However, the reader would be more interested in the average number of samples per station per year.

Agreed. We have summarized this in Table 3.

L188: Can you give some of the published coverage of different hydroclimates from the original CAMELS references? What range do you really cover here?

We added Figure 5 in the previous revision to exactly summarize this concern.

L225: Further above you stated that you had a large number of chemistry samples what do not have a matching instantaneous discharge observation. How do these numbers (here >90%) fit together?

Changed.

L240-245: This section mostly contains data source, interpolation methods, GIS-matching. All this belongs to the method section and not to the results.

Agreed and moved.

Example analyses
L275ff: All this needs a proper method description in the methodology chapter. I strongly miss a proper description of the actual results (only a weak verbal description and a link to Fig. 7) and a scientific interpretation of the results. Just stating that more sophisticated methods may reveal underlying controls is not enough here.

This section was rewritten

L278: Maybe it slipped through my read but what is FDC?
Added in methods.

L303ff: All what I stated for the first example is also valid for the second one.
Agreed and edited.

To Authors:

The CAMELS-Chem dataset will be a useful tool for catchment research, and I look forward to seeing this paper published. What appears to be the most novel and intensive part of this work is the addition of atmospheric deposition inputs to the CAMELS catchments. Mirroring comments from previous reviewers, I think that presenting this work as a data paper is the most appropriate path. As such, I also still have some comments about increasing transparency of the methods.

As someone familiar with accessing large amounts of USGS flow and water quality data from the NWIS, I think that the paper should make it more clear how the dataset they present is not just a compilation of flow and water quality for the same group of catchments. Be very explicit and detailed with what was done for data cleaning, interpolation, gap filling, etc. I find the language about ETL in Section 2.2 confusing, mainly in that the end product that I saw (three CSV files on Hydroshare) seems fairly straightforward, whereas the described end product (a relational database using PostgreSQL) sounds very fancy. I fully acknowledge my ignorance of the ETL methodology, but as someone who would be very excited to use CAMELS-Chem, I'd like this section to paint a picture of what exactly is going into the production of your dataset/database, and what I would be downloading to use it.

Regarding the deposition data, that section is also scant on details. What is the resolution of the input data, and what resolution are you reporting the data? The data was also not available for me to review in your Hydroshare link, so I was unable to comment on its formatting.

Again, from my perspective, doing a quick merge of USGS flow and water quality data would be quite easy for someone using the R package "dataRetrieval", and so I think you need to make clear that your dataset is value-added: mainly from the addition of deposition data, but also through your data processing/cleaning procedures and the construction of the relational database. Make sure to clearly sell CAMELS-Chem to the audience! I look forward to seeing this work progress.

We have worked diligently to address the concerns mentioned by rewriting the methods. We have also changed the introduction to better highlight the value of the dataset.

Comments:

1. Line 18. Maybe in the abstract say how many sites are in CAMELS-Chem? See Comment Line 97.

Added

2. Line 24. "Annual deposition loads and concentrations" are promised, but they are not included in the included database, nor are they mentioned in Methods section. See comment from line 239.

Included in database

3. Line 75. "Thus, we now have There are opportunities". This text should be fixed.

Fixed

4. Line 97. Cite the number of sites from CAMELS, and then CAMELS-Chem (516 sites cited at Line 84)

516 is the correct number

5. Line 101. "most of its sites are drawn come from" USGS HBN. Remove the word "come". Also, how many sites from HBN?

Changed to HCDN to match Newman et al. description.

6. Line 103. How many sites from NWIS or HCDN?

They are not mutually exclusive, but essentially all of them.

7. Line 116. You list the challenges: missing data, mis-matched sample times, inconsistent parameter names, or varying units of measure. Of these, later in the paragraph I only see sample times and units of measure discussed. Did you fix parameter names? Or fill missing data? Please elaborate.

Much more detail was added to this section.

8. Line 127. How do the 506 and 488 become 516 sites? More detail please.

Only 506 have coincident discharge and water chemistry but 516 have water chemistry.

9. Line 130. "impute". Could you use a more colloquial form for this sentence to improve readability?

We clarify that imputing is fixing missing data.

10. Line 131. Forgive my ignorance, but does the Hydroshare upload represent your final data repository, using PostgreSQL? Having looked at your CSV files, which I found well organized (see some minor comments later about the dataset), I'm not sure if there's something I'm missing.

You are correct, but the database is now available for download.

11. Line 148. Could you include the number of catchments and number of measurements for all the parameters, as you've done for Si and DOC, into Table 1?

New Table 3

12. Table 1. Units and abbreviation and USGS code are missing for discharge

Corrected

13. Table 1. In a response to a previous reviewer comment about merging solutes, you justify using unfiltered and filtered parameter codes (e.g. chloride, sulfate, nitrate). This justification needs to be in the manuscript, explaining why there are multiple parameter codes in the last column of Table 1. See also my comment on the database, about my confusion about the column "total_no3" and others.

Corrected with citation for combining constituent codes.

14. Line 225. This sentence needs to be corrected (referring to right-most column or bottom-most row). This does not reflect the information in the table. Should it be second column, and then second row? I might advise moving Q to first row/column before Temp.

Corrected

15. Table 3. As mentioned elsewhere, leading zeros should be present for USGS parameter codes.

Added to database as gauge_id2

16. Line 239. Section 3.5. Atmospheric deposition data. This paragraph should be included under Section 2 Methods. In describing the NADP data, what is the data resolution (temporal, spatial)?

Added to method

17. Table 4. Table 4 should probably be in Methods, rather than Results?

Added to methods as Table 1

18. Figure 1: The inset histograms referenced in the figure caption (and shown in manuscript version 1) are no longer present. They should be added to the figure.

Removed from captions (in Figure 1).

19. Comments on the database. Mirroring previous reviewer comments, the data should be able to stand on their own, and so the following comments are requesting context and explanation to improve readability.

All of the following comments were addressed by a new data release on Hydroshare. We appreciate the reviewers attention to detail.

19.1. The paper discusses atmospheric deposition data (measured concentrations and annual flux estimates), but those data do not appear anywhere in the dataset.

19.2. In the "metrics" file, the following columns have no explanation: q_inst, q_15, q_derived, q_derived_note, q_daily_note, q_daily_cd, q_inst_cd, q_15_cd, measure_unit_code, sample_start_dt, sample_start_time, sample_timestamp, q_inst_ts, q_15_ts, inserted_ts, updated_ts, gauge_id2

19.3. In the "dataset" file, there is a second column "gauge_id2". This column is not described in the "metrics" file, or how it is different from "gauge_id"

19.4. Leading zeros. The gauge IDs should have the proper number of leading zeros. The USGS parameters should also have leading zeros. This applies to all CSV files in your database. This is critical for users not familiar with USGS standards, and might be looking for "940" instead of "00940" (for example).

19.5. The columns total_cl, total_no3, and total_so4 are confusing. My guess is that these columns are filtered and unfiltered values, harmonized? These columns are not mentioned in the manuscript Table 1, nor are the methods for calculating the column values in the manuscript.

---

## Author Response (AR4)

**Editor decision: Publish subject to minor revisions (review by editor)**
by Genevieve Ali
**Public justification (visible to the public if the article is accepted and published)**:
Dear authors: thank you for addressing comments in the last round of review. Both reviewers think that your manuscript toes the line between a data paper and an original research paper, which I believe is what has made the review process a bit challenging. That said, both reviewers believe that the release of the data and accompanying manuscript will be beneficial to the community, and I fully agree. I am returning your manuscript for a range of minor edits suggested by the reviewers. I look forward to receiving the final version of your manuscript.
With best wishes,
Genevieve Ali

We greatly appreciate the comments of the editor and reviewers and their patience in improving the manuscript.  Although minor, the revisions have greatly improved the paper and the data availability.  We address all reviewer comments below in blue text.

**Reviewer 1:**

I found the authors have made significant additions to the manuscript methodology, which was my primary concern. Reading over the response to reviewers, I found that each of my comments was acted upon with additions to the methods. The new additions assuage my concern that this was a "simple" compilation of flow and water quality data. In addition, the additional methods associated with the atmospheric deposition data satisfied my previous comments from the last review.
Author response: Thank you for your time and patience in improving the manuscript.  We have addressed all comments below.

I continue to have comments on the dataset itself.
Reviewer comment: - The previously missing Deposition Data ("DepCon_671_1985_2018.xlsx") does not have an associated documentation file, whereas the file "Camel_Chems_Metrics.xlsx" contains documentation for "Camels_chem_1980_2018.csv". Without documentation, I am unable to determine the associated units of measurement.
Author response: We agree and have provided the file DepCon_metadata.xlsx on Hydroshare.

Reviewer comment: - The files within Catchment Attributes also lack documentation, units, etc., and the "gauge_id2" column continues to be present.
Author response: We agree and have provided the file camels_attributes_v2.0.xlsx on Hydroshare .

Reviewer comment: - If the authors aspire for wide use of their dataset, I suggest they significantly improve the documentation files associated with it. A README file outlining the data structure and attributes would be appropriate.
Author response: We agree and have provided the file camels_chem_readme.txt

Reviewer comment: If I were to suggest a higher standard of replicability, I would encourage the authors to make their methodology fully open source: by publishing the algorithms outlined in bullet points in Section 2.3. I am not judging this submission by that standard, but as a potential user, this could be a way to allow for continued integration of updated USGS data (2018 and beyond).
Author response: We have posted the execution files that were used on Hydroshare, in the directory "Source Data Extraction Code (ETL).

**Reviewer 2:**
Reviewer comment: Line 51: Missing a closing parenthesis ")" following "… Addor et al., (2017)".
Author response: Revised as suggested.

Reviewer comment: 125: Strike "National" from "USGS National NWIS"; it's already represented in the acronym.
Author response: Revised as suggested.

Reviewer comment: 136: Change "obtained NADP" to "obtained from NADP".
Author response: Revised as suggested.

Reviewer comment: 229: It appears that the lowercase references to Figure 3 sub-panels are offset by one; for example SO4 and Ca are subpanels "l" and "m" in Figure 3, not "m" and "n" as represented in the document body. This offset needs to be corrected for this and all following references to Figure 3.
Author response: We checked the figure numbers and letters for subpanels carefully and made the changes as requested.

Reviewer comment: 238: Replace "…temperature is measurement relatively…" with "…temperature is generally measured…"
Author response: Revised as suggested.

Reviewer comment: 252: Cumbersome and repetitive sentence beginning with "The range…". I suggest replacing the entire sentence with "The range of hydrological and meteorological conditions represented is nearly identical between CAMELS and CAMELS-Chem catchments."
Author response: Revised as suggested.

Reviewer comment: 268: replace "…covered of the percent of the FDC during…" with "…percent of the FDC covered…"
Author response: Revised as suggested.

Reviewer comment: 276 and 278: Table 4, not 5.
Author response: Revised as suggested.

Reviewer comment: 283: Table 5 not 4.
Author response: Revised as suggested.

Reviewer comment: 297: Suggest replacing this sentence with "In many cases these patterns are consistent with patterns in stream chemistry; for example, patterns of NO3 deposition (Figure 7c) compare closely with the corresponding pattern in chemistry (Figure 1j)."
Author response: Revised as suggested.

Reviewer comment: 337: You appear to be missing a citation in the sentence "In another example use of CAMELS-Chem, … used DIC stream chemistry…"
Author response: We agree and have revised accordingly. The sentence now reads: "In another example Stewart et al. (2022) used DIC stream chemistry from CAMELS-Chem to show seasonal changes were controlled by $CO_2$ concentration distribution with depth, while long-term DIC concentrations were controlled by climate."

Reviewer comment: 369: Replace "…to develop new hypothesis…" with "…to develop a new hypothesis…" if only one hypothesis, or "…to develop new hypotheses…" if multiple.
Author response: Revised as suggested.

Reviewer comment: 389: Replace "coincident" with "coincidence".
Author response: Revised as suggested.

Reviewer comment: 570 and beyond: several of your references are listed twice.
Author response: We removed duplicate references.